# Coordinated host-pathogen transcriptional dynamics revealed using sorted subpopulations and single macrophages infected with *Candida albicans*

José F. Muñoz [1], Toni Delorey [1,2], Christopher B. Ford[1], Bi Yu Li[1], Dawn A. Thompson[1], Reeta P. Rao [1,2] & Christina A. Cuomo [1]

The outcome of fungal infections depends on interactions with innate immune cells. Within a population of macrophages encountering *Candida albicans*, there are distinct host-pathogen trajectories; however, little is known about the molecular heterogeneity that governs these fates. Here we developed an experimental system to separate interaction stages and single macrophage cells infected with *C. albicans* from uninfected cells and assessed transcriptional variability in the host and fungus. Macrophages displayed an initial up-regulation of pathways involved in phagocytosis and proinflammatory response after *C. albicans* exposure that declined during later time points. Phagocytosed *C. albicans* shifted expression programs to survive the nutrient poor phagosome and remodeled the cell wall. The transcriptomes of single infected macrophages and phagocytosed *C. albicans* displayed a tightly coordinated shift in gene expression co-stages and revealed expression bimodality and differential splicing that may drive infection outcome. This work establishes an approach for studying host-pathogen trajectories to resolve heterogeneity in dynamic populations.

[1] Broad Institute of MIT and Harvard, Cambridge, MA 02142, USA. [2] Worcester Polytechnic Institute, Worcester, MA 01609, USA. These authors contributed equally: José F. Muñoz, Toni Delorey. Correspondence and requests for materials should be addressed to R.P.R. (email: rpr@wpi.edu) or to C.A.C. (email: cuomo@broadinstitute.org)

Interactions between microbial pathogens and the host innate immune system are critical to determining the course of infection. Phagocytic cells, including macrophages and dendritic cells, are key players in the recognition of and response to fungal infections[1]. *Candida albicans*, the most common fungal pathogen, can cause life threatening systemic infections in immunocompromised individuals; however, in healthy individuals, *C. albicans* can be found as a commensal resident of the skin, gastrointestinal system, and urogenital tract[2]. In addition, *C. albicans* can withstand harsh host environments, including the macrophage phagosome, by regulating metabolic and cell morphology pathways[3,4]. While macrophages directly control fungal proliferation and coordinate the response of other immune cells, the outcomes of these interactions are heterogeneous; some *C. albicans* cells are effectively killed by macrophage engulfment whereas others evade or survive macrophage interactions and persist in the host[5].

Previous studies of *C. albicans* and immune cell interactions in bulk populations have identified key pathways by characterization of either the fungal or host transcriptional response during these interactions[3,6,7]. More recently, dual transcriptional profiling of host–fungal pathogen interactions has also examined populations of cells[8–11]. Bulk approaches measure the average transcriptional signal of millions of cells, obscuring differences between infection fates. Even in a clonal population of phagocytes, many immune cells do not engulf any fungal cells, while others can phagocytose up to ten fungal cells[12]. Single-cell RNA sequencing (scRNA-Seq) has highlighted the substantial variation in gene expression between cells within stimulated or infected immune cell populations[13–15]. For example, scRNA-Seq revealed that a subset of macrophages exposed to bacterial stimuli displayed a strong interferon response, which was associated with cell surface variation between different bacteria[14]. A recent study measured host and pathogen gene expression in single host cells infected with the bacteria *Salmonella typhimurium*; however, the low number of pathogen transcripts detected per cell was only sufficient for analysis of sets of co-regulated genes[16]. To date, parallel transcriptional profiling of single host cells and fungal pathogens has not been reported.

To overcome these challenges, here we develop an experimental system to isolate subpopulations of distinct infection outcomes and examine host and pathogen gene expression in sorted subpopulations and in single, infected macrophages. We focus on four distinct infection outcomes: (i) infected macrophages with live *C. albicans*, (ii) infected macrophages with dead, phagocytosed *C. albicans*, (iii) macrophages exposed to *C. albicans* that remained uninfected, and (iv) *C. albicans* exposed to macrophages that remained unengulfed. In addition to carrying out dual RNA-Seq on these subpopulations, we isolate single macrophages infected with *C. albicans* and adapt methods to measure gene expression of the host and pathogen to further resolve heterogeneity. By comparing the transcriptional profiles of *C. albicans* and primary, murine macrophages at both the subpopulation and single infected cell levels, we characterize the tightly coupled time-dependent transcriptional responses of the host and pathogen across distinct infection fates. We establish that both host and pathogen gene expression can be measured from single cells; this reveals that genes involved in host immune response and in fungal morphology and adaptation show expression bimodality or changes in splicing patterns, variation that is important to consider in monitoring the dynamics of host–fungal pathogen interactions.

## Results

### Characterization of heterogeneous macrophage–*Candida* interactions.
To capture infection subpopulations and more finely examine host and pathogen interactions, we developed a system for fluorescent sorting of *C. albicans* with macrophages. We utilized a reporter to measure fungal cell status (live or dead) and infection status (engulfed or unengulfed). This construct, which constitutively expresses green fluorescent protein (GFP) and mCherry, was integrated into *C. albicans* at the *NEUT5* locus (Methods); when *C. albicans* cells lyse in the acidic macrophage phagosome, GFP loses fluorescence upon the change in pH[15], whereas mCherry remains stable for up to 4 h in this environment as visualized by microscopy. Primary, murine bone-derived macrophages were stained with CellMask Deep Red plasma membrane stain. To study host–fungal pathogen infection stages at finer resolution, the *C. albicans* reporter strain was then co-incubated with primary bone-derived, stained macrophages and subpopulations were isolated using fluorescence-activated cell sorting (FACS) at time intervals (0–4 h; Methods; Fig. 1a). At each time point, cells were processed and sorted as rapidly as possible and kept on ice during sorting; however, as transcriptional changes could have occurred during the processing time, we have compared only samples that were similarly processed (Methods). These time points were selected to capture the early transcriptional changes of *C. albicans* in response to interactions with macrophages[3]. To examine gene expression, RNA of both host and fungal cells was extracted and adapted for Illumina sequencing using Smart-Seq2 (Methods). Four major infection subpopulations were isolated by FACS: (i) macrophages infected with live *C. albicans* (GFP+, mCherry+, Deep red+), (ii) macrophages infected that phagocytosed and killed *C. albicans* (GFP−, mCherry+, Deep red+), (iii) macrophages exposed to *C. albicans* (GFP−, mCherry−, Deep red+), and (iv) *C. albicans* exposed to macrophages (GFP+, mCherry+, Deep red−); (Fig. 1a). The number of uninfected macrophages exposed to *C. albicans* ranged from an average of 61 to 67%, while the number of uninternalized *C. albicans* ranged from 22 to 7% over the time course (Fig. 1b). The number of infected macrophages varied from 11 to 29% over the time course; the largest increase in this population was observed between 0 and 1 h postinfection and then remained stable over 2 and 4 h (Fig. 1b). The number of macrophages infected with dead, phagocytosed *C. albicans* ranged between 0 and 3% over the time course (Fig. 1b).

The number of RNA-Seq reads and transcripts detected for both host and fungal pathogen subpopulations was sufficient to profile parallel transcriptional responses (Supplementary Note 1). Based on alignments to a composite reference of both mouse and *C. albicans* transcriptomes (Methods), the fraction of mapped reads for host and pathogen was highly correlated with the size of the transcriptomes and percent of sorted cells for each subpopulation (e.g., 87% host and 13% fungus for macrophages infected with live *C. albicans*; Fig. 1c; Supplementary Figure 1B; Supplementary Data 1). In subpopulations of macrophages infected with live fungus, an average of 10,333 host and 4,567 *C. albicans* genes were detected (at least 1 fragment per replicate across all samples; Supplementary Figure 1A; Supplementary Data 1). We focused the differential expression analysis on subpopulations with high transcriptome coverage and highly correlated biological replicates (e.g., Pearson's $r > 0.85$, and $> 6,000$ and 2,000 transcripts detected in macrophages and *Candida*, respectively; Supplementary Figure 1B). The detection of a high number of transcripts across sorted subpopulations supports that we have established a robust system for defining host and fungal pathogen gene-expression profiles during phagocytosis.

Next, we examined the major expression profiles in both the host and fungus. In *C. albicans* and macrophages, respectively, we identified 588 and 577 differentially expressed genes (DEGs; fold change (FC) $> 4$; false-discovery rate (FDR) $< 0.001$) among all

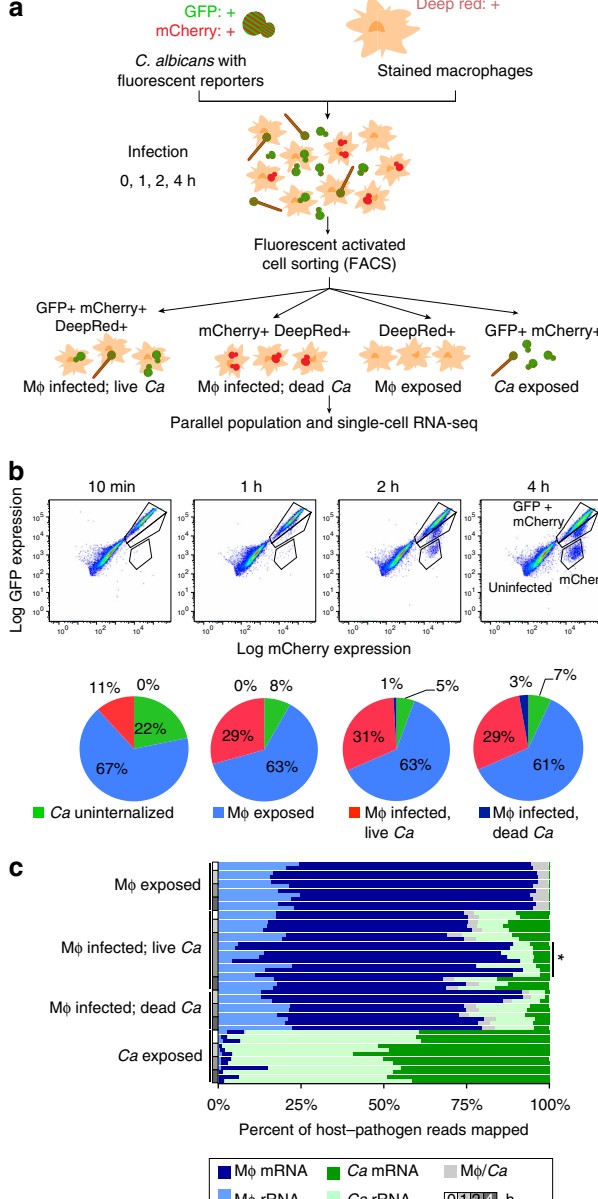

**Fig. 1** Defining macrophage–*Candida* interactions using cell sorting and RNA-Seq. **a** Schematic representation of the experimental model, using primary, bone marrow-derived macrophages (BMDMs) incubated with *Candida albicans* reporter strain CAI4-F2-mCherry-GFP, sampling at time intervals, and sorting to separate subpopulations of interacting cells. **b** Bone marrow-derived mouse macrophages (Mø) were incubated with the *Candida albicans* (*Ca*) reporter strain (Neut5L-*NAT1*-mCherry-GFP), then sorted at 10, 60, 120, and 240 min using fluorescence-activated cell sorting on the BDSORP FACSAria. Pie charts depicting the percent of cells sorted for each infection subpopulation. **c** Percent of host (Mø: macrophages) and pathogen (*Ca*: *C. albicans*) RNA-Seq reads mapped to the composite reference transcriptome, including mouse (GRCm38/mm10; mRNA and rRNA transcripts) and *Candida albicans* (CAI4-F2; mRNA and rRNA transcripts). Mø/*Ca* (gray color label) is the proportion of multiple mapped reads, i.e., reads that map to both the mouse and *C. albicans* transcriptomes, which were excluded from further analysis

pairwise comparisons (Methods; Fig. 2; Supplementary Data 2). To determine the major patterns of infection-fate specific or interaction-time specific, we used DEGs to perform principal component analysis (PCA), then PC scores were clustered by

*k-means* (Methods). PCA revealed that *C. albicans* subpopulations primarily clustered by infection fate (exposed and phagocytosed), with more subtle variation within these populations over time, as described below (Fig. 2a). By contrast, the transcriptional response in infected and exposed macrophages primarily varied over time and were highly similar at each time point between these subpopulations (Fig. 2b). These findings highlight how cell sorting can be used to separate different infection fates and distinguish gene-expression signatures within populations of host and pathogen cells.

**Metabolic and morphological adaptation in phagocytosed *Candida*.** We next examined how *C. albicans* gene expression varied across exposed and phagocytosed cells over time. Using *k-means* clustering of 588 DEGs, we identified 4 clusters of genes with similar expression patterns; the major patterns of expression across time were either induced rapidly upon phagocytosis (clusters 1 and 2) or repressed at 1 and 2 h upon phagocytosis (clusters 3 and 4; Fig. 2c; Supplementary Figure 2A). Comparing the phagocytosed and un-engulfed *C. albicans* subpopulations at each time point, the highest number of DEGs was found at 1 h ($n = 165$), highlighting a rapid and specific transcriptional response upon phagocytosis, which was maintained throughout the 4-h infection time course (Supplementary Data 2; Fig. 2c).

Genes highly induced in phagocytosed *C. albicans* at 2 and 4 h (cluster 1, $n = 93$) are involved in adaptation to the macrophage environment including changes in metabolic pathways (Fig. 2c, e). These genes are involved in glucose and carbohydrate transport, carboxylic acid and organic acid metabolism, and fatty acid catabolic processes (enriched GO-terms corrected $P < 0.05$, hypergeometric distribution with Bonferroni correction; Supplementary Data 3). Prior microarray analysis of bulk populations of *C. albicans* exposed to macrophages reported similar changes[3]; by sorting infection fates, our work suggests that this response is specific to engulfed *C. albicans*, allowing the pathogen to utilize the limited spectrum of nutrients available in the phagosome[3]. We also found that genes involved in glyoxylate metabolism, the beta-oxidation cycle, and transmembrane transport were significantly induced in phagocytosed *C. albicans* relative to exposed cells. Using sorted populations revealed that multiple classes of transporters were highly up-regulated in both engulfed and exposed *C. albicans* subpopulations (including oligopeptide transporters, several high-affinity glucose transporters, and amino acid permeases), suggesting these changes are not in response to phagocytosis (Fig. 2c; Supplementary Datas 2 and 3). Genes most strongly induced at 4 h upon phagocytosis (cluster 1) are involved in pathogenesis and associated with the formation of hyphae, including four of the eight genes involved in the core filamentation response[17] (Supplementary Datas 2 and 3). While media containing serum can also induce *C. albicans* filamentation, we found that filamentation genes were more highly induced in the phagocytosed *C. albicans* subpopulation (Supplementary Data 2; Supplementary Figure 2B). Genes induced most strongly at 1 and 2 h upon phagocytosis (cluster 2) are involved in early fatty acid oxidation response and transmembrane transport (Opt and Hgt classes; Supplementary Datas 2 and 3). These transporters differ from those in cluster 1 in that they show peak expression at 1 h in phagocytosed cells and decrease in expression by 4 h (Fig. 2c). Clusters 1 and 2 contained 6 confirmed or putative transcription factors (*SUT1*, *STP4*, *TEA1*, *ADR1*, *ZCF38*, *TRY4*) (Supplementary Data 2), all of which encode zinc finger containing proteins; zinc cluster transcription factors have been implicated in *C. albicans* virulence[18,19].

Two sets of genes were specifically downregulated in live, phagocytosed *C. albicans* at 1 and 2 h. Expression levels of these

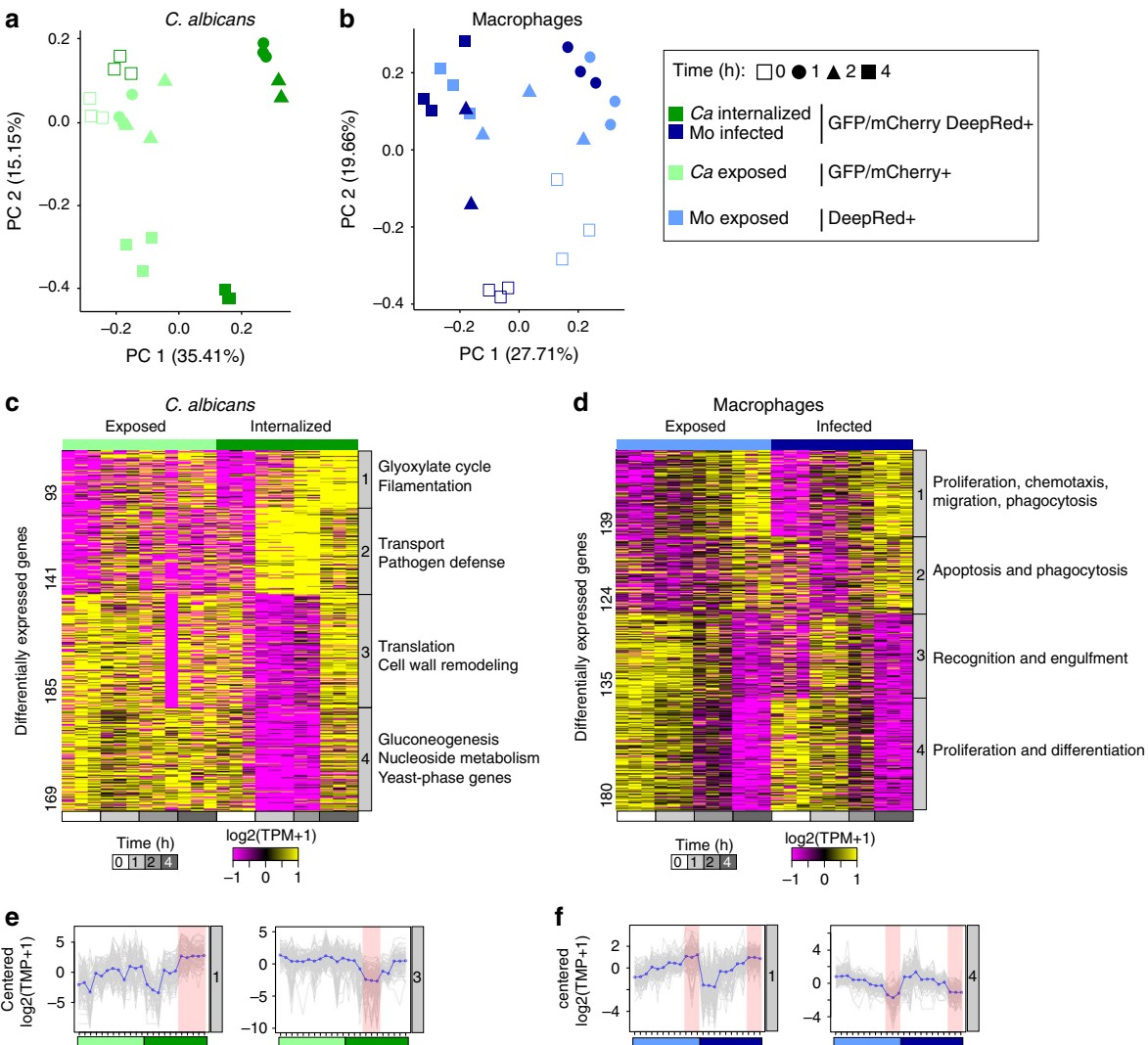

**Fig. 2** Dual RNA-Seq profiling revealed dynamic host–pathogen response. Principal component analysis (PCA) using the transcript abundance for the pathogen (**a**; *Candida albicans*) and the host (**b**; macrophages) using significantly differentially expressed genes (DEGs; FC > 4, FDR < 0.001) in live *C. albicans*-phagocytized macrophages (GFP+ mCherry+ Deep Red+) compared to all other conditions. Contributions of each infection outcome replicate (points) to the first two principal components (PC1 and PC2) are depicted. The projection score (red: high; blue: low) for each gene (row) onto PC1 and PC2 revealed four clusters in both host and pathogen. For each projection score cluster, immune response genes (macrophages) and functional biological categories (*C. albicans*; deducted from GO-term enrichment analysis) are shown. **c** Transcriptional response in subpopulations of *C. albicans*. Heatmap depicts significantly differentially expressed genes (color scheme of gene expression level (log$_2$(transcripts per million (TPM) + 1)) from −1 (purple) to 1 (yellow)) for replicates of each *C. albicans* sorted populations (exposed and phagocytosed) at 0, 1, 2 and 4 h postinfection, grouped by *k-means* (similar expression patterns) in four clusters. Each cluster includes synthesized functional biological relationships using Gene Ontology (GO) terms (corrected *P* < 0.05; Fisher's Exact Test). **d** Transcriptional response in subpopulations of macrophages. Heatmap depicts significantly differentially expressed genes (color scheme of gene expression level (log$_2$(TPM + 1)) from −1 (purple) to 1 (yellow)) for replicates of each macrophage-sorted populations (exposed and infected) at 0, 1, 2 and 4 h postinfection, clustered by *k-means* (similar expression patterns). Each cluster includes synthesized functional biological relationships using IPA terms (−log(*P* value) > 1.3; *z*-score > 2; right-tailed Fisher's exact test). Expression patterns for **e** clusters 1 and 3 in *C. albicans* and **f** cluster 1 and 4 in macrophages. The expression level for each gene is plotted (gray) in addition to the mean expression profile for that cluster (blue)

genes largely did not change in exposed *C. albicans* over the time course (clusters 3 and 4; Fig. 2c, e). Notably, these repressed genes recovered their expression levels by 4 h in phagocytosed *C. albicans*. Cluster 3 included genes related to the translation machinery and peptide biosynthesis, including ribosomal proteins, chaperones, and transcription factors that regulate translation (enriched GO terms, corrected-*P* < 0.05, hypergeometric distribution with Bonferroni correction; Supplementary Data 3; Supplementary Figure 2B). Repression of the translation machinery was previously noted in *C. albicans* during

macrophage interaction[3]. Our results demonstrate that downregulation of ribosomal proteins, chaperones and translation-regulator transcription factors are specific to phagocytosed *C. albicans* and that expression of these genes recovered at later time points (Fig. 2c, e; Supplementary Figure 2B). Cluster 3 also encompassed genes involved in morphological and cell surface remodeling, including an essential negative regulator of filamentation, *SSN6* (Fig. 2c; Supplementary Figure 2B; Supplementary Data 2). Repressed genes in Cluster 4 are largely involved in nucleoside metabolic processes, gluconeogenesis and host

adaptation (enriched GO terms, corrected-$P < 0.05$, hypergeometric distribution with Bonferroni correction; Fig. 2c; Supplementary Data 3). A subset of these repressed genes are yeast-phase specific and included those involved in ergosterol biosynthesis, cell growth, and cell wall synthesis (Fig. 2c; Supplementary Figure 2). These results highlight that a large part of the observed transcriptional repression is specific to phagocytosed *C. albicans*, in contrast to the common sets of upregulated genes shared by exposed and phagocytosed cells.

**Pathogen recognition and pro-inflammatory response to Candida**. In parallel with the analysis of *C. albicans* gene expression, we also examined the transcriptional response of macrophages. Across all samples, we identified 577 DEGs (FC > 4; FDR < 0.001; Supplementary Data 4), which grouped into four clusters with similar expression patterns (Fig. 2b, d). For both exposed and infected macrophages, a major difference was found between 1 and 4 h along PC1, which highlights genes that were highly induced or repressed at 4 h in these subpopulations (clusters 1 and 4, respectively; Fig. 2b, d). These 4 clusters together were significantly enriched for genes involved in activation of phagocytosis, migration of phagocytes, and triggering the innate immune response; this includes the induction of pathways such as IL-6, IL-8, and NF-κB signaling, Fcγ Receptor-mediated phagocytosis, production of nitric oxide and reactive oxygen species (ROS), pattern recognition receptors, RhoA, ILK, and leukocyte extravasation signaling (*P* value < 0.05 right-tailed Fisher's exact test; Fig. 2d; Supplementary Figure 3). While these pathways were activated in both exposed and infected macrophages, pathogen recognition was more highly induced in exposed cells and the production of ROS pathways was more highly induced in infected macrophages (Supplementary Figure 3). Activation of some of these pathways is consistent with previous analysis of phagocyte transcriptional responses to *C. albicans* infection (Supplementary Figure 4)[8–10,20]; sorting distinct infection subpopulations during early time points of infection established that host cells regulate subsets of genes upon *C. albicans* exposure and phagocytosis (Fig. 2d; Supplementary Figure 5).

In exposed and infected macrophages, many of the genes induced at 1 h maintained this expression level at 2 and 4 h (cluster 1; Fig. 2d, f). These genes are related to defense mechanisms such as pro-inflammatory cytokine production and fungal recognition via transmembrane receptors. Upregulated genes related to pro-inflammatory cytokines included tumor necrosis factor (*Tnf*), interferon regulatory factor 1 (*Irf1*), and the chemokine receptor *Cx3cr1*, associated with an innate mechanism of fungal control in a model of systemic candidiasis[21] and colitis[22] (Fig. 2d; Supplementary Figure 5; Supplementary Data 4). A second set of genes was initially repressed at 1 h and then upregulated in both exposed and infected macrophages at 2 and 4 h (cluster 2; Fig. 2d; Supplementary Data 4). This set of genes is associated with pathogen recognition, opsonization, and activation of the engulfment (*P* value < 0.05 right-tailed Fisher's exact test; Supplementary Data 5), including the lectin-like receptor galectin 1 (*Lgals1*), transmembrane receptors (*Fcer1g*), chemokines (*Ccl3*, *Cxcl2*), extracellular complement protein (*C1qb*), and transcriptional regulators that play a role in inflammation and programmed cell death (*Fos*, *Irf8*, *Cebpb*, and *Card9*) (Supplementary Datas 4 and 5). Since the expression of these genes increased at 2 h and maintained high expression in infected macrophages, they may also play an important role during phagocytosis or allow for uptake of additional *C. albicans* cells. The chemokines *Cxcl2*, *Ccl3*, and *Cx3cr1* have also been previously shown to be induced during *C. albicans* interactions with other host cells, including neutrophils in vitro[10], in a murine

kidney model[8], a murine vaginal model[20], and in mouse models of hematogenously disseminated candidiasis and of vulvovaginal candidiasis in humans[9], highlighting the role of these genes in host defense against *C. albicans* infection of different tissues (Supplementary Figure 4).

We also examined subpopulations of macrophages that have phagocytosed and killed *C. albicans;* these data were analyzed separately, as the total number of cells sorted and therefore the transcriptome coverage were low (<3,000 host transcripts detected) and had modestly correlated biological replicates (e.g., Pearson's $r < 0.56$). We found a small set of highly induced pro-inflammatory cytokines, including *Ccl3*, *Cxcl2*, *Il1rn*, and *Tnf*, and transcription regulators such as *Cebpb*, *Irf8*, and *Nfkbia* (Supplementary Figure 6). These genes were also induced in macrophages infected with live *C. albicans* (clusters 1 and 2; Fig. 2d), indicating that maintaining expression of these genes may be important for pathogen clearance and host cell survival after phagocytosis.

Another major shift in macrophage gene expression occurred at 4 h, with sets of genes involved in the immune response highly repressed at this later time point (clusters 3 and 4, respectively; Fig. 2d, f). Repressed genes at 4 h (cluster 3) are enriched in cytokines (*Il1a*, *Cxcr4*) and transmembrane receptors, including intracellular toll-like receptor (*Tlr5*), C-type lectin receptors (*Clec4a3*, *Clec10a*, *Olr1*), and interleukin receptors (*Il1r1*). This cluster was also enriched in categories associated with proliferation and immune cell differentiation (*P* value < 0.05 right-tailed Fisher's exact test; Supplementary Data 5). In addition, highly repressed host genes at 4 h (cluster 4) were enriched for categories related to phagosome formation, phagocytosis signaling, and immune response signaling (*P* value < 0.05 right-tailed Fisher's exact test; Supplementary Data 5). Notably, these repressed genes included several interleukin receptors (*Il21r*, *Il4r*, *Il17ra*) and transmembrane receptors (*Tlr9*, *Mrc1*) that are typically highly expressed during the immune response to fungal infections[8,10]. This suggests that during phagocytosis of *C. albicans* there is a strong shift in macrophages toward a weaker pro-inflammatory transcriptional response by 4 h.

**Detection of host–pathogen gene expression at a single-cell level**. Even in sorted populations, individual cells may not have uniform expression patterns, as, even in a clonal population, cells can follow different trajectories over time. To address this, we next examined the level of single cell transcription variability during these infection time points. We collected sorted, single macrophages infected with live or with dead, phagocytosed *C. albicans* at 2 and 4 h and adapted the RNA of both the host and pathogen for Illumina sequencing using Smart-Seq2 (Fig. 1a; Methods). With this approach, each infected macrophage and the corresponding phagocytosed *C. albicans* received the same sample barcode, allowing us to pair transcriptional information for host and pathogen at the single, infected cell level (Fig. 3a). While we successfully isolated single infected macrophages via FACS, we cannot control for the number of *C. albicans* cells inside of each macrophage with this approach, since macrophages can phagocytose variable numbers of *C. albicans* cells[12]. We obtained 4.03 million paired-end reads per infected cell on average; a total of 449 single, infected macrophages had more than 1 million paired-end reads (Supplementary Data 1). For macrophages with live *C. albicans*, we found an average 75% of reads mapped to host transcripts and 11% of reads mapped to *C. albicans* transcripts (Fig. 3b). Although parallel sequencing of host and pathogen decreases the sensitivity to detect both transcriptomes from a single library, of the 224 single macrophages infected with live *C. albicans* with more than 0.5 million reads, 202 (90.2%) had at

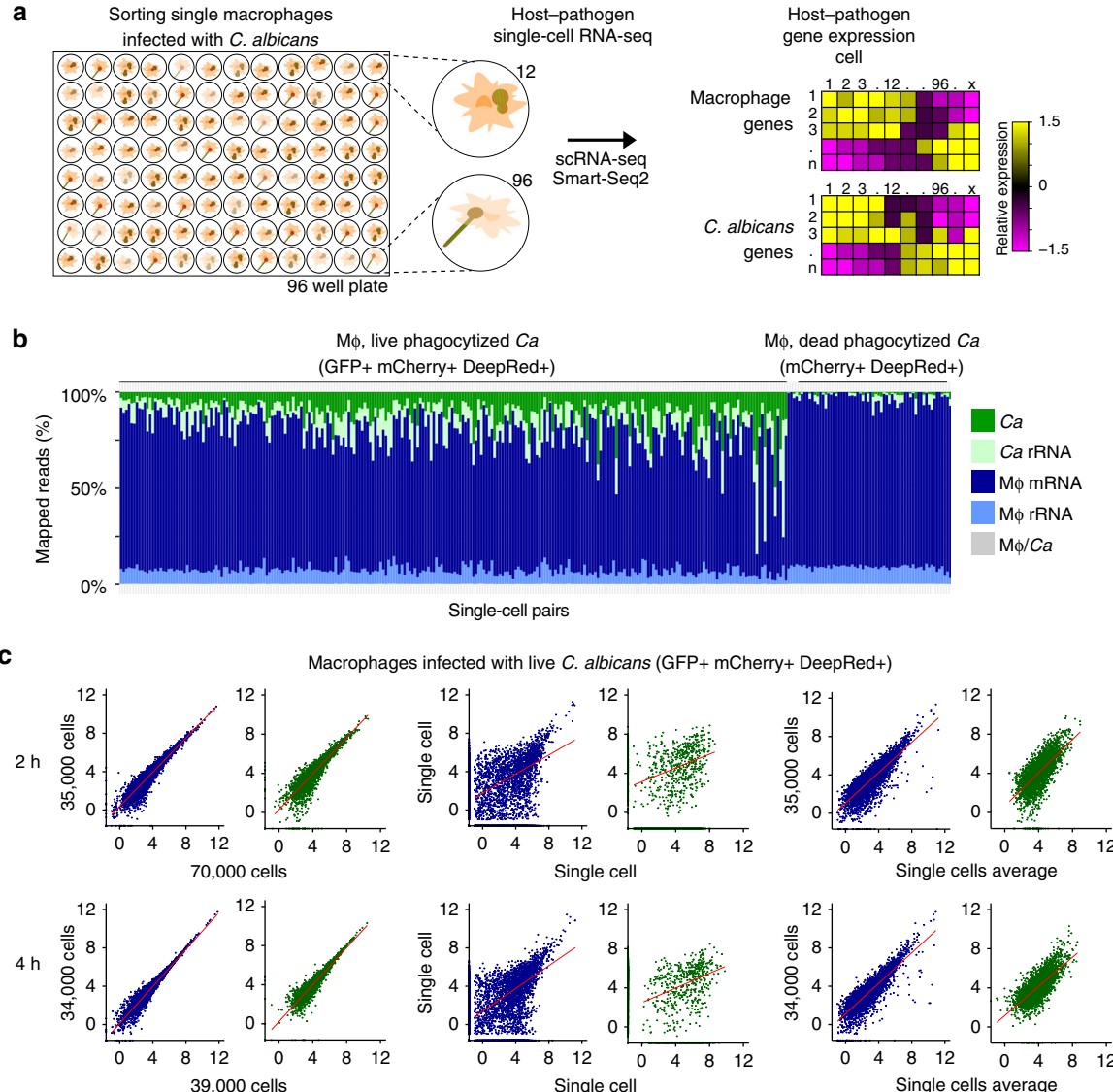

**Fig. 3** Processing and evaluation of parallel host-pathogen single-cell RNA-Seq. **a** Schematic representation of the experimental model for sorting and scRNA-Seq analysis of single macrophages infected with *C. albicans*. In the heatmap, the color scheme of gene expression level (log₂(TPM + 1)) varies from −1.5 (purple) to 1.5 (yellow)) This approach simultaneously produces the transcriptome of both macrophages and *C. albicans* and allows resolve population heterogeneity and trace distinct trajectories during host–pathogen interaction. **b** Plot of the percent of mapped reads to the composite reference transcriptome (mouse messenger and ribosomal RNA + *Candida albicans* messenger and ribosomal RNA collections) of 314 single macrophages with live (mCherry+ GFP+) or with dead, phagocytosed (mCherry+ GFP−) phagocytosed *C. albicans* cells. **c** Plots of the gene expression correlation between (left) two replicates of the sorted subpopulation of macrophages (blue) with phagocytosed live *C. albicans* (green) at 2 and 4 h postinfection; (middle) between two single macrophages and two phagocytosed *C. albicans*, and (right) between the single cells expression average and one replicate of the population

least 2,000 host-transcripts detected (>1 transcripts per million (TPM); 3,904 on average), and 162 (72.3%) had at least 600 *C. albicans* transcripts detected (>1 TPM; 1,435 on average; Fig. 3b; Supplementary Figure 7). The fact that we detected fewer fungal transcripts relative to the host was expected, as the fungal transcriptome is approximately four times smaller than the host transcriptome; this also likely reflects the larger size of the macrophage cells relative to *C. albicans*. Relative to the number of transcripts detected in the RNA-Seq of subpopulations of macrophages infected with live *C. albicans*, in single-infected-cells we detected 38 and 31% (on average) of the transcripts for host and *C. albicans* respectively, indicating that we obtained adequate sequencing coverage for both species. Additionally, we found that

pooling single infected cell expression measurements could recapitulate the corresponding subpopulation expression levels. We found that the extensive cell-to-cell variation between single infected macrophages (average Pearson's from *r* 0.18 to 0.88; Fig. 3c) was reduced when we aggregated the expression of 32 single-cells (Supplementary Figure 8). These results are consistent with previous single-cell studies of immune cells[13,14,23] and indicate that we can accurately detect gene expression in single-infected macrophages and phagocytosed *C. albicans*.

**Dynamic host–pathogen co-stages defined by single-cell analysis.** To finely map the basis of heterogeneous responses during

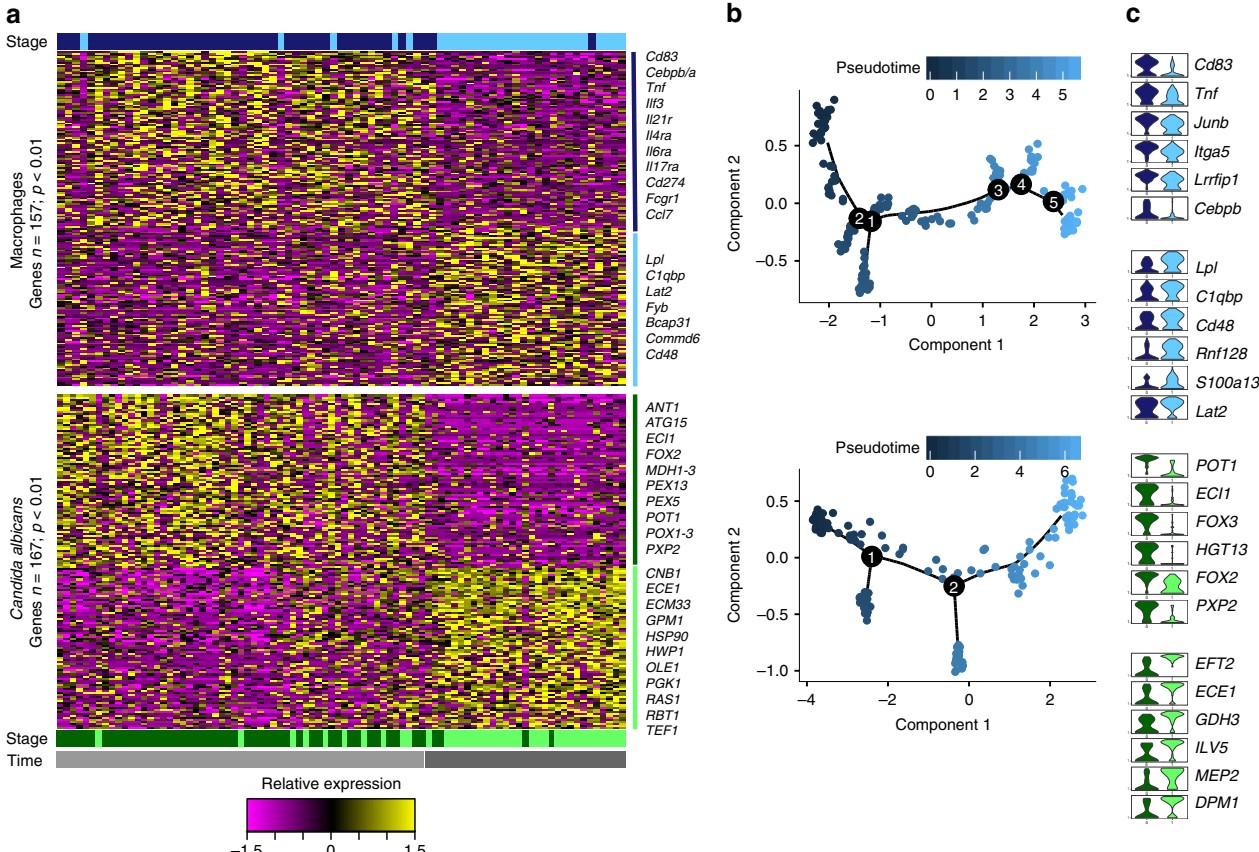

**Fig. 4** Host–pathogen scRNA-Seq profiling traced infection outcome. **a** Heat maps report parallel scaled expression [log TPM (transcripts per million) + 1 values] of differentially expressed genes for each co-state of macrophages infected with live *C. albicans*. Each column represents transcriptional signal from a single, infected macrophage and the *C. albicans* inside of it. (Top: macrophage response; bottom: *C. albicans* response). Color scheme of gene expression is based on *z*-score distribution from −1.5 (purple) to 1.5 (yellow). Bottom and right margin color bars in each heat-map highlight co-state 1 (dark blue in macrophages (1M), and dark green in *C. albicans* (1C)) and co-state 2 (blue in macrophages (2M), and green in *C. albicans* (2C)), and time postinfection 2 h (gray) and 4 h (dark gray). **b** tSNE plot for macrophages (top) and *C. albicans* (bottom) colored by the pseudotime (dark blue to light blue). **c** Violin plots at right illustrate expression distribution of a subset six differentially expressed immune response or immune evasion genes for each co-state in macrophages (dark blue at stage 1; light blue at stage 2) and *C. albicans* (dark green at stage 1; light green at stage 2), respectively

infection, we clustered cells by differential expressed genes and identified host–pathogen co-stages of infection in groups of infected macrophages and phagocytosed *C. albicans* pairs that showed similar gene expression profiles (Figs. 3a and 4a). Briefly, we used genes exhibiting high variability across the infected macrophages and live, phagocytosed *C. albicans* at 2 and 4 h. We then reduced the dimensionality of the expression with PCA, and clustered cells with the t-distributed stochastic neighbor embedding approach (t-SNE)[24] as implemented in Seurat[25] (Methods). The transcriptional response among single-infected macrophages exhibited two time-dependent stages associated with expression shifts from 2 to 4 h (stage 1M and stage 2M; Fig. 4a; Supplementary Figure 9A). While cells were largely separated into these two stages by time, a small subset of macrophages were assigned to the alternate cell stage by unsupervised clustering and appeared to be either early or delayed in the initiation of the transcriptional shift in genes involved in the immune response. DEGs in stage 1M (*n* = 88; likelihood-ratio test (LRT)[26], *P* < 0.001) are related to the pro-inflammatory response, and their expression significantly decreases in stage 2M at 4 h (DEGs *n* = 70; LRT[26], *P* < 0.001; Fig. 4a, top; Supplementary Data 6). This set comprises pro-inflammatory repertoire, such as cytokines (*Tnf, Ilf3, Ccl7*), transmembrane markers (*Cd83, Cd274*), interleukin receptors

(*Il21r, Il4ra, Il6ra, Il17ra*) and the high-affinity receptor for the Fc region (*Fcgr1*), and the transcriptional regulators *Cebpb* and *Cebpa* (Fig. 4a, c). Many genes variably expressed in these single, infected macrophages were not found to be differentially expressed in subpopulations of infected and exposed macrophages at these time points (e.g., *Tnf, Orl1*; Supplementary Figures 5 and 10); this highlights that cell-to-cell variability within each time point observed in single infected macrophages is obscured when surveying populations of cells.

As each single, infected macrophage received a unique sample barcode and host and fungal transcription were measured simultaneously (Fig. 3a), the expression from each macrophage (Fig. 4a, top) was matched with that of the live, phagocytosed fungus (Fig. 4a, bottom). Notably, independent analysis of the parallel fungal transcriptional response identified two pathogen stages that were also primarily distinguished by time. In *C. albicans*, genes significantly upregulated in stage 1C (*n* = 80; LRT[26], *P* < 0.001; Supplementary Data 7) were enriched in organic acid metabolism (*P* = 6.63e−11; enriched GO term, corrected-*P* < 0.05, hypergeometric distribution with Bonferroni correction; Supplementary Data 8), including transporters (*HGT13*) and glyoxylate cycle genes, specifically those from beta-oxidation metabolism (*ECI1, FOX3, FOX2, PXP2*; Fig. 4a, c).

Most macrophages infected by *C. albicans* in stage 1[M] induced a strong pro-inflammatory response (co-stage of infection 1; Fig. 4). At 4 h in stage 2[C], expression of these transporters and glyoxylate cycle genes was reduced instead expression of genes ($n = 86$; LRT[26], $P < 0.001$; Supplementary Data 7) enriched in carbon metabolism was increased of ($P = 2.43e{-}05$; enriched GO term, corrected-$P < 0.05$, hypergeometric distribution with Bonferroni correction; Supplementary Data 8), including genes related to glycolysis and gluconeogenesis (*PGK1*), fatty acid biosynthesis (*FAS1*, *ACC1*), and genes associated with filamentation (*ECE1*, *HWP1*, *OLE1*, *RBT1*); at this stage, the majority of infected macrophages downregulated expression of pro-inflammatory cytokines (co-stage of infection 2; Fig. 4a).

To more finely trace how cells shift their expression program between infection co-stages we performed pseudotime analysis using Monocle[27] (Methods). Briefly, we mapped a trajectory of possible transitions states between the two co-stages using the expression profiles of macrophages or *C. albicans* cells. We observed that the majority of infected macrophages and phagocytosed *C. albicans* pairs followed a linear expression trajectory between two major endpoints or clusters, recapitulating the two major infection co-stages; some cells exhibited alternative expression paths, suggesting the existence of minor cell trajectories (Fig. 4b; Supplementary Figure 11). This ordering of cells revealed a group of infected macrophages at 2 h that expressed high levels of pro-inflammatory cytokines that decreased over time across intermediate and late groups (Fig. 4b, top; Supplementary Figure 11). Similarly, in phagocytosed *C. albicans*, an group of cells at 2 h showed low-expression levels of genes related to filamentation which increased over time (Fig. 4b, bottom; Supplementary Figure 11). In this analysis, *C. albicans* appeared to have an earlier transition to the second co-state; the fraction of phagocytosed *C. albicans* shifting to high expression of filamentation genes across pseudotime increased slightly faster than the fraction of macrophages exhibiting decreased levels of expression of pro-inflammatory cytokines (10% more cells in the late pseudotime range; Fig. 4b). This suggests that the expression heterogeneity in macrophages could be driven by the expression heterogeneity in *C. albicans* (Fig. 4b). This was also supported by unsupervised clustering, where some phagocytosed *C. albicans* collected at 2 h were assigned to the second cell stage and appeared to initiate the transcriptional shift in immune response earlier than in the corresponding macrophage (Fig. 4a). In summary, the major co-stages of infection largely correspond to time of infection; however, trajectory analysis highlights an asynchronous and linear transition associated with the induction of filamentation and metabolic adaptation in *C. albicans* which, correlates with a shift from a strong to a weak pro-inflammatory gene expression profile in the host.

**Expression bimodality in single macrophages infected with Candida.** To further examine heterogeneity in gene expression, we characterized modality of expression profiles across host-pathogen single cells, and then compared these distributions between 2 and 4 h using a normal mixture model and Bayesian modeling framework as implemented in scDD[28] (Methods). Overall, an average of 84.5% of genes detected across single-infected macrophages displayed unimodal gene-expression distributions (Fig. 5a; Supplementary Data 9). Highly expressed unimodal genes (top 5%) with similar expression levels at 2 and 4 h encompassed genes involved in opsonization and *C. albicans* recognition, such as complement proteins (*C1qb*, *C1qc*) and galectin receptors that recognize beta-mannans (*Lgals1*, *Lgals3*). In phagocytosed *C. albicans*, an average of 76% had unimodal

expression patterns (Fig. 5a; Supplementary Data 9). Highly expressed unimodal genes (top 5%) with similar expression levels in both co-stages were enriched in the oxidation–reduction process and defense against ROS (enriched GO term, corrected-$P < 0.05$, hypergeometric distribution with Bonferroni correction; Supplementary Data 9). These unimodal genes highlight the core genes involved in the host immune response to fungus and pathogen virulence, respectively.

A subset of the genes highly expressed in co-stages of single-infected macrophages and phagocytosed *C. albicans* showed bimodal expression distributions (Figs. 4c and 5). As bimodal transcriptional heterogeneity among single stimulated or infected cells can signify distinct immune cell expression programs[13,23], we next examined whether subgroups of macrophages could be defined by shared bimodality of genes involved in the immune response. An average of 15% host genes showed patterns of bimodal expression among or within 2 and 4 h (exceeded bimodality index threshold; Dirichlet process mixture of normals model; Supplementary Data 9, Fig. 5a). In addition to expression bimodality, some of these genes showed patterns of differential distributions (e.g., shifts in mean(s) expression, modality, and proportions of cells) across and within 2 and 4 h as implemented in scDD package[28]. This includes genes involved in pathogen intracellular recognition and pro-inflammatory response that had a bimodal expression distribution at 2 h but not at 4 h (e.g., *Olr1*, *Tnfrsf12a*), bimodal expression only at 4 h (*Il4ra*), or bimodal expression at 2 and 4 h with differential mean expression (*Il21r*, *Il17ra*; $P < 0.05$, Benjamini–Hochberg adjusted Fisher's combined test; Fig. 5b; Supplementary Data 9). As observed in macrophages infected with *Salmonella*[14] or stimulated with LPS[14], *Tnf* and *Il4ra* also exhibited bimodal expression patterns in single macrophages infected with *C. albicans* (Fig. 5b, top). In addition, we found unique subsets of genes displaying differential distributions in single macrophages infected with *C. albicans*, but not in response to bacterial stimuli, including *Il17ra* and other lectin-like receptors (*Olr1*; Fig. 5b; Supplementary Figure 12). This suggests that variably expressed pathogen-specific receptors may play a role in these interactions, even in clonal populations of cells.

We next characterized variation in isoform usage between single macrophages during *C. albicans* infection, including immune response genes. Briefly, we calculated the frequency (percentage spliced in) of previously annotated splicing events and identified differential isoform usage between single-infected macrophages using BRIE[29] (Methods). We detected differential splicing between macrophages in 144 genes, including the immune response genes *Clec4n* (*Dectin-2*), *Il10rb* and *Ifi16* (cell pairs > 2000; Bayes factor > 200; Supplementary Data 10). Notably, Dectin-2 had differential exon retention between macrophage stages, with the Dectin-2 α isoform (6 exons) predominantly in stage 1, and Dectin-2 β isoform (5 exons) predominately in stage 2 (Fig. 6). The truncated isoform Dectin-2 β lacks part of the intracellular domain and most of the transmembrane domain of the receptor[30]. We found that Dectin-2 β has a lower posterior probability of transmembrane helix ($P = 0.80$; TMHMM2) as compared as Dectin-2 α ($P = 0.99$; TMHMM2; Supplementary Figure 13). The lack of this transmembrane region has been proposed to encode a secreted protein, which may act as an antagonist to full-length Dectin-2[30]. Activation of Dectin-2 receptors on macrophages and dendritic cells by *C. albicans* leads to Th17 T-cell differentiation to assist in the immune response[31,32]. These results suggest that splicing variation among single macrophages might indicate different potentials to respond to fungal infection.

We next hypothesized that *C. albicans* may also demonstrate expression heterogeneity and bimodality that is linked with

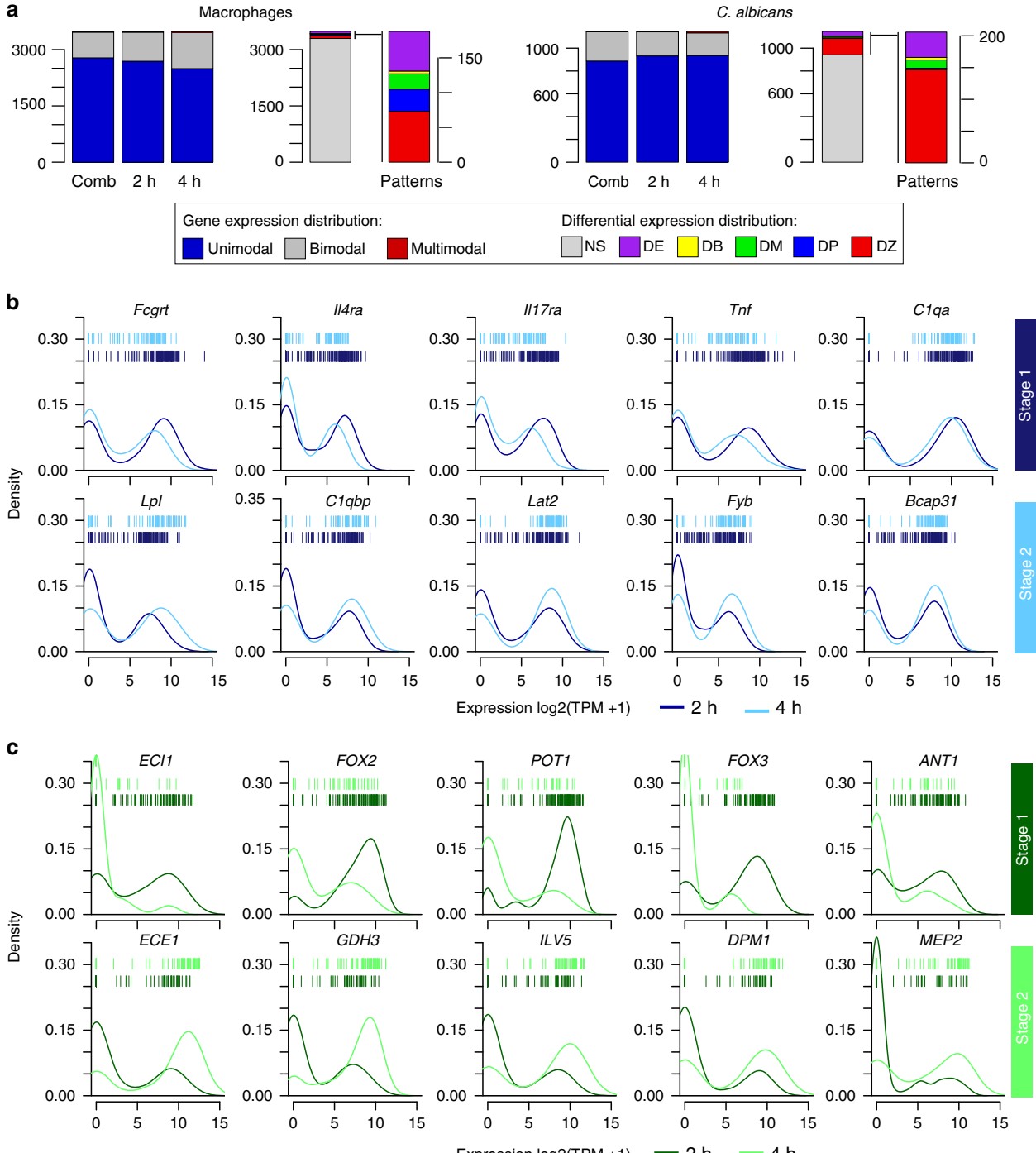

**Fig. 5** Expression variability and bimodality at the single-cell level. **a** Number of genes categorized as unimodal, bimodal, or multimodal (>2 components) according genes expression patterns in single, infected macrophages and corresponding phagocytized *C. albicans* across cells from 2 and 4 h, and within each time point. Differential distributions were assigned as differential expression of unimodal genes (DE), differential modality and different component means (DB), differential modality (DM), differential proportion for bimodal genes (DP), differential proportion of zeroes (DZ), and not significant (NS). **b** Expression density distributions in parallel infected macrophages (n = 267; blue) and **c** phagocytized live *C. albicans* (n = 215; green) for 5 top marker genes in co-state1 and co-state2 across macrophages–*C. albicans* single cells at 2 and 4 h postinfection. Individual cells are plotted as bars for 2 h (top row) and 4 h (bottom row) for each distribution

expression in the corresponding macrophage cell. We found an average of 23% *C. albicans* genes that showed patterns of bimodality at 2 and 4 h, including a subset of virulence-associated genes that also showed shifts in mean expression, modality, and proportions of cells across and within 2 and 4 h (P < 0.05, Benjamini–Hochberg adjusted Fisher's combined test; Fig. 5a; Supplementary Data 11). Differential expression of those genes explained most of the variation across infection co-stages (Fig. 4a)

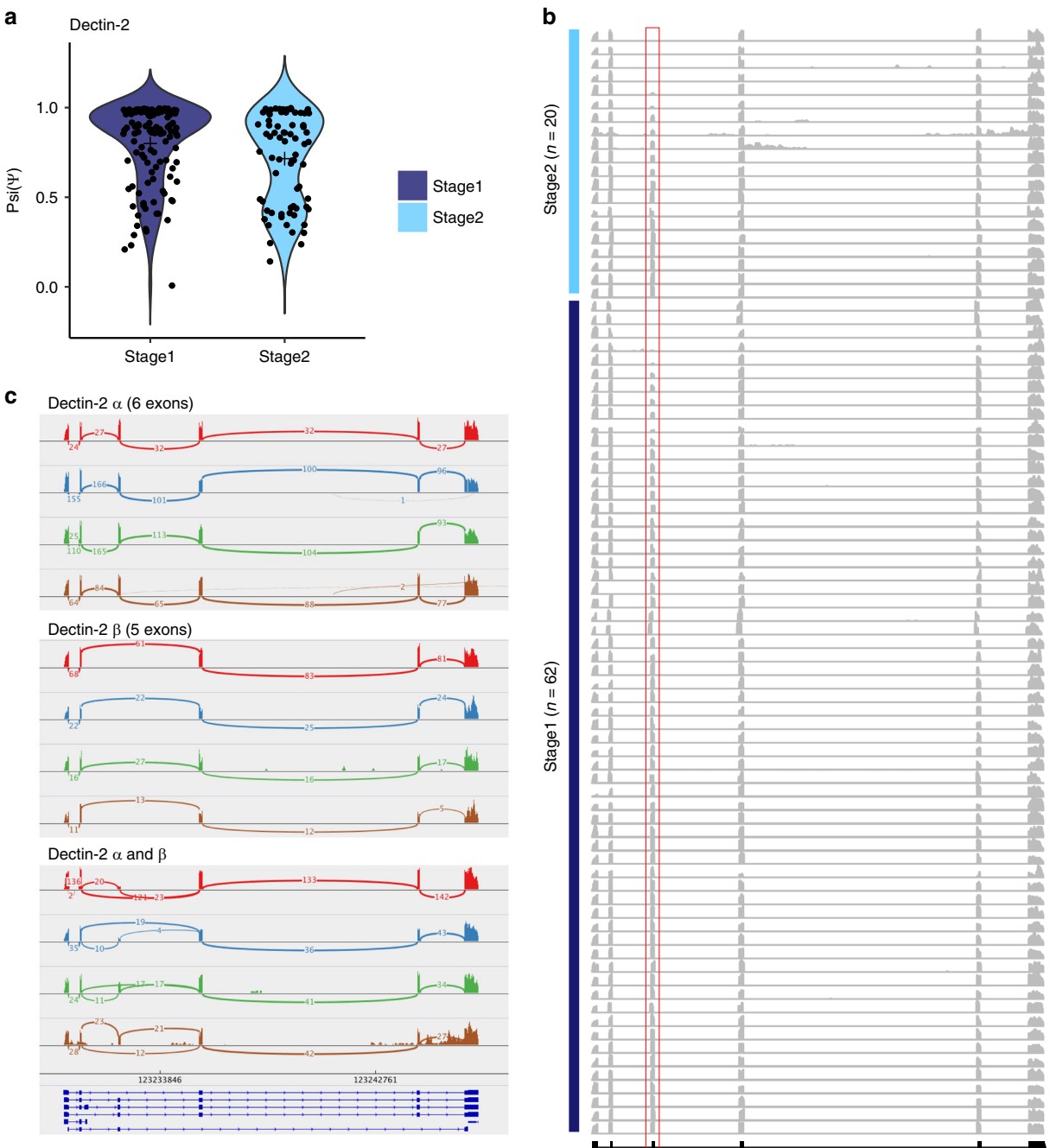

**Fig. 6** Differential splicing and isoform usage in single infected macrophages. **a** Differential inclusion of exon 3 alternative splicing event in Clec4n (Dectin-2) in single macrophages infected with *C. albicans*. Violin plots depict the distribution of the percent spliced-in (Psi/Ψ) scores for single macrophages in stage 1 and 2 of infection. **b** scRNA-seq densities for all exons of Dectin-2 across 82 single macrophages infected with *C. albicans*; 62 in stage 1 and 20 in stage 2. Each row represents a single-cell. Red box indicates the spliced exon 3. **c** Sashimi plots showing differential splicing between single macrophages, including the read densities and number of junction reads. Four cells were selected showing three scenarios: top: Dectin-2 α, middle Dectin-2 β, and bottom cell with both isoforms

and were more important for *C. albicans* fate decision and trajectory (Fig. 4b). This set of genes were enriched in cell adhesion and filamentation, oxidation–reduction process and fatty acid oxidation (enriched GO term, corrected-$P < 0.05$, hypergeometric distribution with Bonferroni correction; Supplementary Data 11), including genes involved in the core filamentation network (*ALS3, ECE1, HGT2, HWP1, IHD1, OLE1*), beta-oxidation and glyoxylate cycle (*ANT1, MDH1-3,*

*FOX2, POX1-3, PEX5, POT1*), and response to oxidative stress (*DUR1,2, GLN1, PGK1*; Supplementary Data 11; Fig. 5c). Measuring dual species gene expression in sorted infection subpopulations and in single infected cells reveals that expression heterogeneity and bimodality of genes involved in fungal morphology and adaptation are tightly coregulated. We observed shifts in the host response during host–fungal pathogen interactions, and, in some cases, this might result in different

expression levels of host immune response genes and pathogen virulence genes. This approach can further enhance our understanding of distinct infection fates and the correlated gene regulation that governs host cells and fungal pathogen interaction outcomes.

## Discussion

Host and fungal cell interactions are heterogeneous, even within clonal populations. One way to resolve this heterogeneity is to subdivide these populations by infection fate or stage to measure gene expression variability across subpopulations. Single-cell approaches offer the ultimate level of subdivision to study heterogeneity and may obviate the need for sorting when run at sufficient scale. Both these approaches rely on measuring host and fungal pathogen gene-expression levels using dual RNA-sequencing to provide insight as to how both species respond in each infection stage. Here, we piloted both approaches to study fungal interactions with host cells. We developed a generalizable strategy to isolate distinct host and fungal pathogen infection fates over time, including single-infected cells. In both sorted populations and single-infected cells, we demonstrated that gene-expression changes can be measured in both host and pathogen simultaneously. This approach allowed characterization of distinct infection fates within heterogeneous host and fungal pathogen interactions. This approach could be used to better further characterize the requirement for specific host and pathogen genes for these infection responses, and single-cell analysis is well suited to characterize variability in both the host and pathogen during active infections.

This dual scRNA-Seq approach builds on prior transcriptional studies of interactions between microbial pathogens and immune cells. For fungi, RNA-Seq studies have largely measured gene-expression profiles of either the host or the fungal pathogen[3,6,7] and have measured transcription profiles across infection outcomes[8–11]. To better examine heterogeneity of host–pathogen interactions, recent approaches demonstrated the use of a similar GFP live/dead reporter to measure gene expression in phagocytes infected by bacterial pathogens using scRNA-Seq[14–16]. However, these studies mainly focused on the transcriptional response of the host, as bacterial transcriptomes can be difficult to measure due to their relatively low number of transcripts[14,16]. By contrast, we have demonstrated that both host and *C. albicans* gene expression can be measured in single infected cells; further studies will be needed to examine if the same or modified approaches can be extended to other microbial–host interactions.

By examining single-infected macrophages, we showed that host and pathogen exhibit transcriptional co-stages that are tightly coupled during an infection time course, providing a high-resolution view of host–fungal interactions. This also revealed that expression heterogeneity of key genes in both infected macrophages and in phagocytosed *C. albicans* may contribute to infection outcomes. We identified two, time dependent linear co-stages of host-fungal pathogen interaction, with potential intermediate stages in some single cells suggested by pseudotime analysis. The initial co-stage is characterized by induction of a pro-inflammatory host profile after 2 h of interaction with *C. albicans* that then decreased by 4 h. This is consistent with studies in human macrophages, where pro-inflammatory macrophages that interact with *C. albicans* for 8 h or longer skew toward an anti-inflammatory proteomic profile[33]. Both commensal and invasive stages of *C. albicans* infection are impacted by the balance between pro-inflammatory and anti-inflammatory responses[34]. In single macrophages infected with *C. albicans*, the shift to an anti-inflammatory state, including upregulation of genes involved in the activation of inflammasomes, was coupled with

the activation of filamentation and cell-wall remodeling in *C. albicans*. Previous work has shown that *C. albicans* can switch from yeast to hyphal growth within the nutrient-deprived, acidic phagosome[35] and can escape by rupturing the macrophage membrane during intraphagocytic hyphal growth[36]. Other mechanisms of escape include the activation of macrophage programmed cell death pathways, including the formation of inflammasomes and pyroptosis[37], or cell damage induced by a cytolytic peptide toxin (Candidalysin *ECE1*)[38]. Our analysis of single cells revealed bimodal expression in genes involved in these processes, and time analysis suggests that an initial shift in expression of filamentation and cell-wall remodeling programs in phagocytosed *C. albicans* rapidly result in downregulation of the pro-inflammatory state of the host cells. This supports the hypothesis that an asynchronous *C. albicans* yeast-to-hyphae transition within the macrophage could drive expression heterogeneity in the host. Alternatively, the downregulation observed in the macrophages after this time point may not be directly related to the fungal state and expression, and instead may reflect self-regulation of the duration of the pro-inflammatory response. While gene-expression patterns are tightly correlated in host and fungal pathogen and the progression in co-stages of infection appears largely linear, greater time resolution across more diverse cell populations will be needed to fully resolve the expression heterogeneity among individual infected macrophages and phagocytosed *C. albicans*. Further work using this approach could also examine the role of specific genes in both *C. albicans* and the host to clarify the drivers of transcriptional responses and heterogeneity in host–fungal pathogen infection fates.

We found that both single-infected macrophages and the corresponding phagocytosed *C. albicans* cells exhibit expression bimodality for a subset of genes. The expression bimodality observed for the host as well as the pathogen is consistent with the evolutionary concept of bet hedging[39]. Both cell types may rely on stochastic diversification of phenotypes to improve their survival rate in the event of an encounter with the other cell type. For instance, clonal populations of *C. albicans* that find themselves in the unpredictable, changing environment of a host phagocyte could increase the chance of survival by varying the expression of key genes involved in the response. A similar scenario may provide an advantage for phagocytes that encounter pleomorphic *C. albicans*. These strategies are noted to occur within microbial populations, where a small fraction of "persister" cells might be capable of surviving exposure to lethal doses of antimicrobial drugs as a bet-hedging strategy[40]. Larger numbers of host-pathogen pairs collected over a longer time course could disambiguate how expression co-stages in *C. albicans* and macrophages results in distinct infection fates.

Heterogenous transcriptional responses are important to consider in the treatment of fungal infections. Genes expressed uniformly among fungal cells in a population may be more effective therapeutic targets than the products of genes expressed by only a subset of cells. A comparison of transcriptional signal of drug-treated single infected cells to nontreated cells would determine if all phagocytosed *C. albicans* cells coregulate uniformly regulate genes involved in the response to drug treatment or if there is evidence of bimodal expression for genes involved in drug targets or efflux pumps, for example. In addition, single-cell analysis revealed bimodal expression and differential splicing of *Clec4n* (*Dectin-2*), a C-type lectin receptor that recognizes diverse fungal pathogens, as well as some bacteria and parasites[41]. The switch from producing a full length isoform to one missing an exon involved in signaling is related to the shift to an anti-inflammatory state. Parallel host–fungal pathogen expression profiling at single-cell level could not only allow researchers to pinpoint changes in immune recognition and response pathways

but also measure the dynamics of other stimuli, for example, to characterize how new drug treatments affect both pathogen and host cells. While we used sorting to enrich for cell populations of interest, as scRNA-Seq microfluidic platforms continue to develop and profiling thousands of single cells becomes more cost effective, it will become tractable to interrogate more complex samples, including those containing multiple host cell types and nonclonal pathogens. These data will provide fine mapping of diverse microbial immune responses and communication between immune cells. Importantly, this approach will allow further investigation into how fungal phenotypic and expression heterogeneity drives host responses and provide a systems view of these interactions.

## Methods

**C. albicans reporter strain construction.** The reporter construct used in this study was prepared by integrating the GFP and mCherry fluorescent tags driven by the bidirectional *ADH1* promoter and a nourseothricin resistance (NAT[R]) cassette at the Neut5L locus of *C. albicans* strain CAI4-F2[42]. CAI4 is a derivative of SC5314, with genotype ura3::imm434/ura3::imm434 iro1/iro1::imm434[43,44]. The CAI4-F2 strain used to create the CAI4-F2-Neut5L-NAT1-mCherry-GFP reporter strain was a gift from the Fink lab (Whitehead Institute, Cambridge, MA). CAI4-F2 harbors a homozygous *URA3* deletion which makes it less filamentous than other common laboratory strains of *C. albicans*, including SC5314 (Supplementary Figure 14); this allowed more *C. albicans* cells to be sorted (see Macrophage and *C. albicans* infection assay, below). Briefly, the pUC57 vector containing mCherry driven by *ADH1* promoter (Bio Basic) was digested and this portion of the plasmid was ligated into a pDUP3 vector[45] containing GFP, also driven by *ADH1* promoter, a NAT[R] marker, and homology to the Neut5L locus. The resulting plasmid was linearized and introduced via homologous recombination into a neutral genomic locus, Neut5L, using chemical transformation protocol with lithium acetate. Transformation was confirmed via colony PCR and whole-genome sequencing. A whole-genome library was created from strain CAI4-F2-Neut5L-*NAT1*-mCherry-GFP using Nextera-XT library construction strategy (Illumina) and sequenced on an Illumina MiSeq (150 × 150 paired end sequencing; approximately 28 million reads). Sequencing reads were de novo assembled using dipSPAdes[46]. Scaffolds were queried back to the plasmid sequence used to transform CAI4-F2 using BLAST[47]. Sequencing coverage was visualized using integrative genomics viewer[48].

**Macrophage and C. albicans infection assay.** Primary, bone-derived macrophages (BMDMs) were derived from bone marrow cells collected from the femur and tibia of C57BL/6, female mice. All mouse work was performed in accordance with all relevant ethical regulations for animal testing and research, under a Broad Institute and Massachusetts Institute of Technology Institutional Animal Care and Use Committee (IACUC) approved protocol (0615-058-1). Primary bone marrow cells were grown in "C10" media[49]. C10 media is composed of RPMI 1640 media (no phenol red, no glutamine, Life Technologies), 10% fetal calf serum, 100 units/ml penicillin/streptomycin, 2 mM L-glutamine (ThermoFisher Scientific), 55 uM ß-mercaptoethanol (ThermoFisher Scientific), 0.1 mM MEM nonessential amino acids (VWR), 1 mM sodium pyruvate (VWR), and 10 mM HEPES (VWR). The C10 media was supplemented with macrophage colony stimulating factor (M-CSF) (ThermoFisher Scientific) at final concentration of 10 ng/ml, to promote macrophage differentiation and supplemented with M-CSF (ThermoFisher Scientific) at final concentration of 10 ng/ml, to promote differentiation into macrophages. C10 media was not supplemented with uridine; this led to reduced CAI4-F2 filamentation (compared to wild-type SC5314) and allowed more *C. albicans* cells to be sorted. Cultures were then stained with F4/80 (Biolegend) to ensure that ~95% of the culture had differentiated into macrophages. For the infection RNA-sequencing experiment, BMDMs were seeded in six-well plates (Falcon). Two days prior to the start of the infection experiment, yeast strains were revived on rich media plates. One day prior to the infection experiment, yeast were grown in 3 ml overnight cultures in rich media at 30 °C. On the day of the infection experiment, macrophages were stained with CellMask Deep Red plasma membrane stain (diluted 1:1000) (ThermoFisher Scientific). Macrophages and stain were incubated at 37 °C for 10 min, then macrophages were washed twice in 1× PBS. Two hour prior to infection, yeast cells were acclimated to macrophage media (RPMI 1640 no phenol red, plus glutamine, ThermoFisher Scientific) at 37 °C prior to the infection. Yeast cells were then counted and seeded in a ratio of 1 *C. albicans* cell to two macrophage cells. Yeast and macrophages were then coincubated at 37 °C (5% CO_2). At each selected time point (0 min, 1 h, 2 h, or 4 h), media was removed via aspiration, 1 ml of 1× TrypLE, no phenol red (ThermoFisher Scientific) was added to each well and incubated for 10 min. After vigorous manual pipetting, two wells for each time point were combined into one tube. Each time point was run in biological triplicate. Samples were then spun down at 37 °C, 300*g* for 10 min and resuspended in 1 ml PBS + 2% FCS and placed on ice until FACS. RNAlater was used shortly after sorting, as this reagent leads to decreased GFP expression[50]. To control for gene expression changes that may be induced by FACS or other sample

processing steps (i.e., 10 min incubation with TrypLE and 10-min centrifugation), only sorted samples were used in comparative analyses. We estimate that the sorting time for each sample was between 1 and 5 min, depending on the abundance of each subpopulation being isolated at each time point. Samples were kept on ice before and after sorting and a cold block was used to hold sample tubes during the sort. We estimate that for each time point assayed, an hour passed between the initiation of sorting and flash freezing.

**Fluorescence-activated cell sorting.** Samples were sorted on the BDSORP FACSAria running the BD FACSDIVA8.0 software into 1× PBS and then frozen at −80 °C until RNA extraction. Cells were sorted into the following subpopulations: (i) macrophages infected with live *C. albicans* (GFP+, mCherry+, Deep red+), (ii) macrophages infected with dead, phagocytosed *C. albicans* (GFP−, mCherry+, Deep red+), (iii) macrophages exposed to *C. albicans* (GFP−, mCherry−, Deep red+), and (iv) *C. albicans* exposed to macrophages (GFP+, mCherry+, Deep red−). The strategy used for cell sorting is depicted in Supplementary Figure 15. Single cells were sorted into 5 μl of RLT 1% β-mercaptoethanol in a 96-well plate (Eppendorf) and frozen at −80 °C.

**RNA extraction and evaluation of quality.** RNA was extracted from population samples using the Qiagen RNeasy mini kit. All samples were resuspended in RLT (Qiagen) + 1% β-Mercaptoethanol (Sigma) and subjected to 3 min of bead beating with 0.5 mm zirconia glass beads (BioSpec Products) in a bead mill. Single macrophages infected with *C. albicans* were directly with dead, phagocytosed by sorting cells into a 96-well plate containing 5 μl of RLT (Qiagen) + 1% β-Mercaptoethanol (Sigma).

**cDNA synthesis and library construction.** For population samples, the RT was carried out as described[51], starting with the addition of 2 μl 100 μM Oligo-dT30VN (5′-AAGCAGTGGTATCAACGCAGAGTACT30VN-3′), 2 μl of dNTP mix (10 mM each) and 0.2 μl of RNAse inhibitor (40U/μl, ThermoFisher)) to each sample. Samples were then incubated at 72 °C for 3 min. Whole-transcriptome amplification (WTA) was carried out with the addition of 0.2 μl of Maxima Reverse Transcriptase (Life Technologies), 4 μl of 5× Maxima RT buffer, 2 μl of 10 μM TSO (5′-AAGCAGTGGTATCAACGCAGAGTACATrGrG+G-3′), 1.8 μl of MgCl_2 (100 mM), 0.5 μl of RNase inhibitor (40 U/μl ThermoFisher), 1.5 μl of H_2O and 4 μl of Betaine (5 M) to each sample. Samples were then incubated as follows: 42° for 90 min, 10 cycles of (50 °C for 2 min, 42 °C for 2 min), and heat inactivation at 70 °C for 15 min. Eleven microlitre of the WTA reaction was then used for PCR pre-amplification. PCR pre-amplification was carried out with addition of 12.5 μl of 2× KAPA HiFi HotStart ReadyMix (KAPA Biosystems), 0.5 μl 10 μM ISPCR primer (5′-AAGCAGTGGTATCAACGCAGAGT-3′) and 1 μl H_2O to each sample. Samples were then amplified as follows: 98 °C for 3 min, 14 cycles of 98 °C for 15 s, 67 °C for 20 s, 72 °C for 6 min, with a final extension at 72 °C for 5 min followed by a 4 °C hold.

For single-infected cells, an RNA lysate clean-up was performed with RNAClean XP beads (Agencourt) prior to reverse transcription (RT). RT was carried out as previously described cDNA was generated from single cells based on the Smart-seq2 method as described previously[51]; 0.1 μl 100 μM Oligo-dT (5′-AAGCAGTGGTATCAACGCAGAGTACT30VN-3′), 1.0 μl of dNTP mix (10 mM each), 0.1 μl of RNase inhibitor (40 U/μl; ThermoFisher) and 2.8 μl of trehalose (1 M) were added to each sample. Samples were then incubated 72 °C for 3 min and then placed on ice. WTA was carried out with the addition of 0.1 μl of Maxima Reverse Transcriptase (Life Technologies), 2 μl of 5× Maxima RT buffer, 0.1 μl of 100 μM TSO (5′-AAGCAGTGGTATCAACGCAGAGTACATrGrG+G-3′), 0.1 μl of MgCl_2 (1 M), 0.25 ul with the addition of RNase inhibitor was used at (40 U/ul; ThermoFisher) and 3.4 μl of trehalose (1 M) to each sample. Samples were then incubated at 50 °C for 90 min and 85 °C for 5 min followed by a 4 °C hold. The entire WTA sample then underwent PCR pre-amplification. PCR pre-amplification was carried out with the addition of 12.5 μl of 2× KAPA HiFi HotStart ReadyMix (KAPA Biosystems), 0.5 μl 10 μM ISPCR primer (5′-AAGCAGTGGTATCAAC GCAGAGT-3′) and 2 μl H_2O to each sample. Samples were then amplified as follows: 98 °C for 3 min, 25 cycles of 98 °C for 15 s, 67 °C for 20 s, 72 °C for 6 min, with a final extension at 72 °C for 5 min followed by a 4 °C hold. For both population and single-infected cell libraries, pre-amplification products were purified with AMPure XP beads (Agencourt), quantified with Qubit dsDNA HS Assay Kit (ThermoFisher) and fragment size was assessed with a High-Sensitivity Bioanalyzer Chip (Agilent). Samples were normalized to 0.15–0.20 ng/μl prior to library construction. Library construction was carried out using the Nextera XT DNA Sample Kit (Illumina). Libraries were pooled in an equal molar ratio prior to sequencing. Infection subpopulation samples were sequenced on an Illumina NextSeq (37 × 38 cycles). *Candida* only samples were sequenced on an Illumina MiSeq (75 × 75 cycles). Single infected cells were sequenced on an Illumina NextSeq (75 × 75 cycles). All primers used in this study were synthesized by Integrated DNA Technologies, Inc. (IDT).

**RNA-Seq quantification and analysis for sorted populations.** Basic quality assessment of Illumina reads and sample demultiplexing was done with Picard version 1.107 and Trimmomatic[52]. Samples profiling exclusively the mouse

transcriptional response were aligned to the mouse transcriptome generated from the v. Dec. 2011 GRCm38/mm10 and a collection of mouse rRNA sequences from the UCSC genome website[53]. Samples profiling exclusively the yeast transcriptional response were aligned to transcripts from the *C. albicans* reference genome SC5314 version A21-s02-m09-r10 downloaded from the *Candida* Genome Database (http://www.candidagenome.org).

Samples from the infection assay profiling in parallel host and fungal transcriptomes, were aligned to a "composite transcriptome" made by combining the mouse transcriptome described above and the *C. albicans* transcriptome described above. To evaluate read mappings, BWA aln (BWA version 0.7.10-r789, http://bio-bwa.sourceforge.net/)[54] was used to align reads, and the "XA tag" was used for read enumeration and separation of host and pathogen sequenced reads. Multi-reads (reads that aligned to both host and pathogen transcripts) were discarded, representing only an average of 2.6% of the sequenced reads. Then, each host or pathogen sample file were aligned to its corresponding reference using Bowtie2[55] and RSEM (RNA-Seq by expectation maximization; v.1.2.21). Transcript abundance was estimated using TPM. Since parallel sequencing of host and pathogen from single macrophages increased the complexity of transcripts measured compared to studies of only host cells alone, we detected lower number of transcripts of macrophages as compared with other studies using phagocytes and similar scRNA-seq methods[14,15,23].

**Differential gene expression analysis of population RNA-Seq**. TMM-normalized "transcripts per million transcripts" (TPM) for each transcript were calculated, and differentially expressed transcripts were identified using edgeR[56], all as implemented in the Trinity package version 2.1.1[57]. Genes were considered differentially expressed if they had a 4-FC difference in TPM values and a false discovery rate below or equal to 0.001 (FDR < 0.001), unless specified otherwise.

**Single-cell RNA-Seq quantification and analysis**. BAM files were converted to merged, demultiplexed FASTQ format using the Illumina Bcl2Fastq software package v2.17.1.14. For the RNA-Seq of sorted population samples, paired-end reads were mapped to the mouse transcriptome (GRCm38/mm10) or to the *C. albicans* transcriptome strain SC5314 version A21-s02-m09-r10 using Bowtie2[55] and RSEM (RNA-Seq by expectation maximization; v.1.2.21)[58]. Transcript abundance was estimated using TPM. For read mapping counts paired-reads were aligned to the "composite reference" as described above.

For each single macrophage infected with *C. albicans*, we quantified the number of genes for which at least one read was mapped (TPM > 1). We filtered out low-quality macrophage or *C. albicans* cells from our data set based on a threshold for the number of genes detected (a minimum of 2,000 unique genes per cell for macrophages, and 600 unique genes per cell for *C. albicans*, and focused on those single-infected macrophages that have a high number of transcripts detected in both host and pathogen (Figure S9A). For a given sample, we defined the filtered gene set as the genes that have an expression level exceeding 10 TPM in at least 20% of the cells. After cell and gene filtering procedures, the expression matrix included 3,254 transcripts for the macrophages and 915 transcripts for *C. albicans*. To estimate the number of *C. albicans* in each macrophage, we measured the correlation of GFP levels from FACS with the total number of transcripts detected in live, phagocytosed *C. albicans* cells (at least 1 TPM), but found only a modest correlation between these two metrics ($R^2 = 0.52$).

To eliminate the nonbiological associations of the samples based on plate-based processing and amplification, single-cell expression matrices were log-transformed ($\log(TPM + 1)$) for all downstream analyses, most of which were performed using the R software package Seurat (https://github.com/satijalab/seurat). In addition, we do not find substantial differences in the number of sequenced reads and detected genes between samples. We separately analyzed two comparisons of macrophages–*C. albicans* cells: (i) single macrophages infected—live phagocytosed *C. albicans* cells at 2 and 4 h (macrophages = 267; *C. albicans* cells = 215; macrophage–*C. albicans* = 156); and (ii) macrophages infected and live or dead, phagocytosed *C. albicans* cells at 4 h (macrophages = 142; *C. albicans* cells = 71). These numbers of macrophages and *C. albicans* cells are the total that met the described QC filters.

**Detection of variation across single, infected cells**. To examine if cell to cell variability existed across a wide range of population expression levels, we analyzed the variation and the intensity of non-unimodal distribution for each gene across single macrophages and *C. albicans* cells. Briefly, we determined the distribution of the average expression ($\mu$), and the dispersion of expression ($\sigma^2$; normalized coefficient of variation (CV)), placing each gene into bins, and then calculating a z-score for dispersion within each bin to control for the relationship between variability and average expression as implemented in the R package Seurat[25].

**Detection of variable genes and cell clustering**. To classify the single cell RNA-Seq from macrophages and *C. albicans*, the R package Seurat version 2.0 was used[25]. We first selected variable genes by fitting a generalized linear model to the relationship between the squared CV and the mean expression level in log/log space and selecting genes that significantly deviated (Bonferroni corrected P value < 0.05) from the fitted curve, as implemented in Seurat using the *FindVariableGenes* function (parameters: mean.function = ExpMean, dispersion.function = LogVMR)[15]. Then

highly variable genes (CV > 1.25; Bonferroni-corrected P value < 0.05) were used for PCA, and statistically significant determined for each PC using the JackStraw plot function. Significant PCs (jackstraw adjusted association test P value < 0.05) were used for two-dimension tSNE to define subgroups of cells we denominated as host–pathogen co-states. We identified DEGs (corrected P < 0.05, likelihood ration test) between co-states using a LRT for single-cell differential expression[26] as implemented in Seurat. To perform pseudotime analysis, we applied the Monocle 2.8.0 method[27] using filtered normalized genes from Seurat. We reordered cells in pseudotime along a trajectory using the top 50 DEGs and identified genes with most significant changes as a function of progress along the trajectory using the branched expression analysis modeling (BEAM test; q value < 1e−4).

**Detection of differential expression distributions and bimodality**. To detect which genes have different expression distributions in single macrophages infected with live *C. albicans*, we compared the distributions of gene expression across and between 2 and 4 h and identified genes showing evidence of differential distribution using a Bayesian modeling framework as implemented in scDD[28]. We used the permutation test of the Bayes Factor for independence of condition membership with clustering (*n* permutations = 1000), and tested for a difference in the proportion of zeroes (testZeroes = TRUE). A gene was considered differentially distributed using Benjamini–Hochberg adjusted P values test (P value < 0.05).

**Detection of differential splicing**. To detect alternative splicing and determine variation in isoform usage between single macrophages during *C. albicans* infection, we calculated exon splicing rates in individual macrophages using the data-dependent module of BRIE v0.2.0[29]. BRIE calls splicing at predefined cassette exons and quantifies splicing using exon reads in single-cell data. We used the processed annotation file and sequence features for mouse transcripts (mm10 GRCm38.p5, ANNO_FILE = SE.gold, FACTOR_FILE = mouse_factors.SE.gold). Differential splicing was performed using the brie-diff module for all pairwise comparisons, and genes with differential splicing between single macrophages were defined as cell pairs > 2000 and Bayes factor > 200.

**Functional biological enrichment analyses**. For *C. albicans*, Gene ontology (GO)-term analysis was performed through the *Candida* Genome Database GO Term Finder and GO Slim Mapper (http://www.candidagenome.org[59]). GO terms were considered significantly enriched in a cluster or set of genes if we found a GO term corrected P value lower than 0.05 using hypergeometric distribution with Bonferroni correction.

For macrophages, ingenuity pathway analysis (IPA) was performed. We investigated biological relationships, canonical pathways and Upstream Regulator analyses as part of the IPA software. This allowed us to assess the overlap between significantly DEGs and an extensively curated database of target genes for each of several hundred known regulatory proteins. Clusters or set of genes were considered significantly enriched if we found a −log(P value) greater than 1.3 (i.e., P value ≤ 0.05; right-tailed Fisher's exact test) and z-score greater than 2 as recommended by the IPA software.

**Availability of biological materials**. The CAI4-F2-Neut5L-NAT1-mCherry-GFP reporter strain is available from the ATCC Biodefense and Emerging Infections Research Resources Repository (NR-51634), from the Fungal Genetics Stock Center (FGSC#26694), or upon request to RPR.

**Reporting Summary**. Further information on experimental design is available in the Nature Research Reporting Summary linked to this article.

## Data availability
All sequence data for this project has been deposited in the SRA under Bioproject PRJNA437988. Raw and processed data for gene expression analysis was deposited in the GEO under GSE111731. Whole-genome sequence data of CAI4-F2-Neut5L-*NAT1*-mCherry-GFP has been deposited at the NCBI SRA under SRX4924342.

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

## Acknowledgements

We thank Aviv Regev and members of her lab for providing support for all the experimental work in this paper and Noam Shoresh for helpful discussions about single-cell analysis. We also thank Raktima Raychowdhury for help with preparation of the

primary BMDM cells and Anh Hoang, Mehment Toner, and Daniel Irima for providing microwells used in preliminary experiments. This project has been funded in whole or in part with Federal funds from the National Institute of Allergy and Infectious Diseases, National Institutes of Health, Department of Health and Human Services, under award U19AI110818 to the Broad Institute. CBF was supported by a Helen Hay Whitney postdoctoral fellowship, TD was supported by NIAID and WPI.

## Author contributions

CBF, DAT, RPR, and CAC designed the study. TD, CBF and BYL carried out experiments. JFM analyzed the data and prepared figures and tables. JFM, TD, RPR and CAC wrote the initial draft of the manuscript, which was revised with input from all authors. All authors read and approved the final manuscript.

## Additional information

**Competing interests:** The authors declare no competing interests.

