## [Peer Review File · Nature Communications]

Reviewers' comments:

Reviewer #1 (Remarks to the Author):

This manuscript describes the first study to analyze both sides of a fungal pathogen-host interaction using single cell RNA-seq. Overall, this is a well written manuscript that clearly lays out the merits of performing this type of analysis. For the most part, the experimental design is excellent (see below) and computational analyses are top-notch. The authors provide a potentially very useful dataset for the *C. albicans* community to form hypotheses and test them with mutant strains and cell lines.

Major comments:

I only have one major criticism of the manuscript and it has to do with the fact that the uninfected macrophages and unexposed *C. albicans* were not sorted and it appears (based on the methods section) that RNA-later was not added to the sorted cells until after sorting. If this is true, the "unexposed" sample can't be reasonably compared to the other samples because you are not taking into account any transcriptional changes that might result from potential stress imposed by putting the cells through the FACS machine. I do not think that this totally negates the novelty of this study or the interesting findings, but I do think that the authors should redo the analysis presented in Figure 2 after excluding the "unexposed" samples. I expect that this might alter the exact results but the overall finding might be the same. Either way, this re- analysis is necessary.

Minor Comments:

The phrase "macrophages infected with dead *C. albicans*" is a bit misleading. It implies that you are exposing the macrophages to heat-killed fungi when in reality, the *C. albicans* cells have died following phagocytosis. Please change the phrase in the text and figures to indicate this.

One question that still remains is whether or not this single-cell RNA-seq method (which is much more expensive and more arduous, both computationally and experimentally) will provide biological insight that one can't get from performing RNA-seq on an infected monolayer. Unfortunately, there does not appear to be an equivalent RNA-seq dataset available on monolayers of primary BMDMs infected with *C. albicans*. Is this correct? If this dataset does exist, the authors should do a careful comparison to establish that single-cell RNA-seq in this context is worth all of the effort. If possible, this would greatly strengthen the manuscript.

Reviewer #2 (Remarks to the Author):

Coordinated host-pathogen transcriptional dynamics revealed using sorted subpopulations and single, *Candida albicans* infected macrophages

Authors:

José F. Muñoz, Toni Delorey, Christopher B. Ford, Bi Yu Li, Dawn A. Thompson, Reeta P. Rao, and Christina A. Cuomo

Comments

General

The paper describes an interesting and challenging approach to investigate interactions of subpopulations of macrophages and *Candida albicans* cells on the transcriptional level.

Similar studies have been applied for interactions of bacteria and phagocytes and I completely agree with the authors that the next level of transcriptional profiling studies to study host-pathogen interactions would be to not only looking at the average response of whole populations, but to dissect the response of the different individual subpopulations. From this point of view, the topic is at the forefront of science and of interest for the scientific community.

However, the reason why such studies have yet rarely been published is the fact that it is a very challenging approach with several technical hurdles.

While the expertise of the authors in bioinformatics is obvious and the analysis of the data technical well done, there are a number of critical issues.

(1) A key critical point for all studies of such kind is the cell harvest and RNA extraction protocol. In the current study, the authors harvested the macrophages with/without *C. albicans* cells and the different *C. albicans* reporter strains by centrifugation at 37°C for 10 min, then resuspending the samples in PBS/FCS and placing on ice until FACS. The time required for FACS sorting and the conditions for FACS sorting are not defined. Only after that, the samples are frozen until RNA extraction.

I have strong doubts that the final RNA populations reflect the original transcriptomes of macrophages and *C. albicans* in their culture medium. Previous studies have shown that fungal cells can change their entire transcriptome within less than 10 min and it is likely that transcriptomes of both macrophages and *C. albicans* have changed during the procedure described above, including centrifugation and FACS sorting. I strongly believe that an efficient blockage of transcription within a few minutes (eg by freezing or chemicals such as RNALater) is essential. Controls are required to show that the sampling and harvesting protocols do not influence gene expression profiles for a number of representative genes.

(2) I miss clear quantitative data about the dynamics of interactions: at which time point are how many *C. albicans* cells in contact with macrophages, how many are phagocytosed over the time course, how many fungi are killed, how many produce hyphae, how many escape, how many macrophages are killed etc. I think such a detailed description of the interaction dynamics is essential to understand the profiles and the biological processes in the experimental setting used by the authors (even if that has been done in other studies). This could also show whether the *C. albicans* cells were all in the same stage when exposed to macrophages after their 37°C incubation step, which is implied when the authors attribute the observed differences on the single-cell level to direct host-fungal cell interactions.

(3) Although differences between populations are technically described in detail, the key messages are missing, except that different groups of genes linked to different GO terms are differentially expressed. What I found lacking is which new insights we gained from applying this sophisticated technique that we could not obtain by population-based approaches. This is reflected by the rather short discussion, which is also largely technical in nature.

(4) As such, the study remains descriptive and no clear biological hypothesis has been developed which could be experimentally tested.

Further points:

Line 131: "Fewer transcripts were detected in subpopulations of macrophages infected with dead *C. albicans* (an average of 3,214 host transcripts and 983 fungal transcripts Figure S3A) and...". How meaningful are transcripts of killed *C. albicans* cells?

Reviewer #3 (Remarks to the Author):

The fundamental question under assessment is an important one – how much variation is there in the interaction between a host immune cell and a pathogen, in this case the opportunistic fungal pathogen *C. albicans*, and bone marrow-derived macrophages? The approach taken, that of doing RNA seq in isolated cells undergoing the interaction is clearly the next step in an ongoing refinement in assessing the answer to the question, and the authors are clearly competent in the analysis.

I have a number of comments

The abstract is very long for an abstract – efforts should be made to shorten it, focusing on what is really new in the present study and not on results that support previous work.

The introduction does a pretty good job of defining the field and previous work. Remove the comma on line 77.

The ability to sort the interaction sets is critical to the study – I would like to see the S2A data presented in the manuscript and not in the supplementary data.

In lines 124/125 it is claimed that increases in numbers of engulfed *Candida* cells represents further engulfment. Is it possible that some of the increase represents *Candida* cells proliferating in the engulfed state – and if so do these cells represent yet another possible class of host-pathogen interactions?

In lines 130/131 the authors note that the class of macrophage that have engulfed and potentially killed the *Candida* cells provided very poor data, and are essentially dropped from the analysis, but no explanation is provided. This is a clearly a central class to examine for the study and needs a better fate. Why should this data set be poor? This is a critical point – the authors should either provide this data or provide a good explanation for the failure.

Line 152 - whereas as one word

The initial studies represent analyses of pooled samples selected by the FACS sorting, and these results fundamentally support previous studies directed at the pathogen and host cell responses. Lines 181-185 provide evidence for an up-regulated set that is not attributed to phagocytosis – providing a possible signal here would be useful.

Lines 189/190 suggest that it is the presence of macrophages and not just the medium that is triggering hyphal development, but this claim is poorly supported.

Line 193 – what explains a class of genes that are repressed by interaction and induced by phagocytosis? This is an interesting (confusing) behaviour and deserves a comment.

Next the authors investigate single cell response. This is the critical new area for the paper – what is found in single cells that improves our understanding derived from pooled cells? A first point made is that individual cells have more variation than seen in the previous multi-cell analyses, and then the authors note that pooling the data from the individual cells reduces variation. This seems self evident – what could prevent the pooled samples reducing variation?

The key data that come from the ability to analyze individual cells is the suggestion that there is a coupled response in the cells – a bimodal relationship in the pathogen linked to a bimodal response in the immune cells. In particular – induction of hyphae is coupled to down-regulation of the pro-inflammatory state of the phagocytes. This is an exciting observation, but needs some context. The authors themselves note that the driver of this correlation is not known “it is unclear whether expression heterogeneity among individual infected macrophages results from or results in expression

bimodality among phagocytosed *C. albicans*." Perhaps doing the experiment with a non-hyphal *C. albicans* mutant would clarify this.

Point by point response

We appreciate the detailed comments of all three reviewers and below detail how we have addressed each point. In this revision, we also need to note that the reporter strain used for our analyses was previously mislabeled as SC5314 and more accurately describes as the SC5314 derivative CAI4-F2. This is important to note as we chose to construct the *C. albicans* reporter in SC5314-CAI4-F2 as it is less filamentous than SC5314 when grown in media that does not contain uridine, allowing us to more easily sort *C. albicans* cells. We noted in the text that further work will be needed to optimize our cell sorting approach with highly filamentous *C. albicans* isolates (addressed in **Methods section, Pg. 16; Line 508-510**).

Reviewer #1 (Remarks to the Author):

This manuscript describes the first study to analyze both sides of a fungal pathogen-host interaction using single cell RNA-seq. Overall, this is a well written manuscript that clearly lays out the merits of performing this type of analysis. For the most part, the experimental design is excellent (see below) and computational analyses are top-notch. The authors provide a potentially very useful dataset for the C. albicans community to form hypotheses and test them with mutant strains and cell lines.

Major comments:

I only have one major criticism of the manuscript and it has to do with the fact that the uninfected macrophages and unexposed C. albicans were not sorted and it appears (based on the methods section) that RNA-later was not added to the sorted cells until after sorting. If this is true, the "unexposed" sample can't be reasonably compared to the other samples because you are not taking into account any transcriptional changes that might result from potential stress imposed by putting the cells through the FACS machine. I do not think that this totally negates the novelty of this study or the interesting findings, but I do think that the authors should redo the analysis presented in Figure 2 after excluding the "unexposed" samples. I expect that this might alter the exact results but the overall finding might be the same. Either way, this re- analysis is necessary.

Response: We did not add RNAlater to cells prior to FACS to avoid quenching the GFP signal used for sorting (for example, see Zaitoun et al. *BMC Res. Notes* 2010). To reduce the possibility of transcriptional changes not reflective of the experimental conditions, our samples were kept on ice before and after sorting and samples were flash frozen shortly after collection; processing was done as rapidly as possible and RNAlater was added immediately after sorting. We have included these details in the Methods (**Line 545**). However, the reviewer raises a valid point about potential differences between sorted and unsorted samples affecting our results. To address this, we have now compared sorted samples at time 0 with unsorted samples at time 0. We found there was no difference in *C. albicans* samples and only 21 genes were induced in sorted macrophages relative to the unsorted control. While this suggests that FACS sorting does not had a large impact on the gene expression, we have re-analyzed the differential expression between subpopulations using only sorted samples to mitigate the reviewer's concern. While some genes were no longer detected as differentially expressed in this re-analysis, the overall immunological signatures and clusters of expression patterns remained largely the same (**revised Figure 2**). We have included this re-analysis in this revision by updating the results and replaced **Figure 2** and **supplementary tables 2, 3, and 4** with new versions using the re-analyzed data (**see sections "Subpopulations of phagocytosed C. albicans" Pg. 4; Lines 131-193, and "Subpopulations of macrophages" Pg. 6; Lines 188-244**).

Minor Comments:

The phrase "macrophages infected with dead *C. albicans*" is a bit misleading. It implies that you are exposing the macrophages to heat-killed fungi when in reality, the *C. albicans* cells have died following phagocytosis. Please change the phrase in the text and figures to indicate this.

Response: In the revised manuscript, the phrase "macrophages infected with dead *C. albicans*" has been replaced with "macrophages that have phagocytosed and killed *C. albicans*" throughout the text.

One question that still remains is whether or not this single-cell RNA-seq method (which is much more expensive and more arduous, both computationally and experimentally) will provide biological insight that one can't get from performing RNA-seq on an infected monolayer. Unfortunately, there does not appear to be an equivalent RNA-seq dataset available on monolayers of primary BMDMs infected with *C. albicans*. Is this correct? If this dataset does exist, the authors should do a careful comparison to establish that single-cell RNA-seq in this context is worth all of the effort. If possible, this would greatly strengthen the manuscript.

Response: This single cell analysis has provided high-resolution transcriptional detail of *Candida*-macrophages interaction. As each single infected macrophage was sorted into one well of a plate and all of the transcripts in one well received a unique molecular barcode, we are able to pair the transcriptional signal from a single macrophage with the transcriptional signal of the *Candida* it has engulfed for the first time. We have carried out additional analysis to show that this data can resolve the progression of infection states even when individual cells progress asynchronously, and further supported this by the incorporation of pseudo-time analysis (**revised Figure 4B, described below**). Using this approach, we detected transcriptional bimodality in host and pathogen, a key variable that could contribute to phenotypic diversity that we observe during *in vitro* infection (**see sections "Dynamic host-pathogen co-stages" Pg. 9; Line 276, and "Expression bimodality" Pg. 11; Line 341**). We have also included a new figure for analysis previously presented that summarizes single-cell analysis resolves expression variability and bimodality (**revised Figure 5**). Furthermore, we have added a new figure to highlight the advantages and applications of this single-cell approach (**revised Figure 3A; Line 254**).

To address reviewer's question about biological insights that can be drawn from dual scRNA-Seq, we have performed two new analyses. First, we examined how infection progresses across co-stages using pseudo-time analysis (**revised Methods; Pg. 20; Line 666**). We found that while co-stages of infection largely correspond to the two time points of infection that we sampled, the single cell analysis highlighted an asynchronous and linear transition between these two co-stages; intriguingly, this pseudo-time analysis suggests that this transition appears to be driven by earlier transcriptional changes in the fungal pathogen, followed by a rapid changes in the host transcription (**Pg. 10; Line 319; new Figure 4C, new supplementary figure S11**). Second, we have examined the level of alternative splicing of macrophage transcripts between single cells across infection co-stages. We found variation in transcript isoforms between single macrophages during *C. albicans* infection (**see revised Methods; Pg. 21; Line 681**), including for immune response genes *Clec6a* (*Dectin-2*), *Il10rb* and *Ifi16*. In addition, we found differential exon retention in *Dectin-2* between macrophages in early and late infection stages; macrophages at 2 hours predominantly expressed the α isoform of *Dectin-2* whereas macrophages at 4 hours predominantly expressed the β isoform (**Pg. 12; Line 378; new Figure 6 and supplementary table S13**). These analyses demonstrate how dual scRNA-Seq has uncovered patterns of regulation not detected in previous bulk infection studies.

As dual scRNA-Seq is relatively new, there is no available dataset of primary BMDMs infected with *C. albicans* that would could use for comparisons. However, to contrast our findings with bulk approaches and *C. albicans*/macrophages transcriptional responses, we had compared our subpopulation RNA-Seq dataset with bulk RNA-Seq experiments for either the host response or *C. albicans* response (see *C. albicans* response (**see sections "Subpopulations of phagocytosed *C. albicans*" Pg. 4; Lines 144, 152, 157, 171; and "Subpopulations of macrophages" Pg. 6; Lines 197, 205, 217, 240; Figure S4**)). While the overall response is similar to prior microarray analysis of *C. albicans*

exposed to macrophages, by sorting infection fates, we determined the specific genes in the phagocytosed *C. albicans* subpopulation. Importantly, our dual approach provide new insight on how host cells regulate subsets of genes upon *C. albicans* exposure and these expression patterns are tightly correlated. This could not been detected in previous work as they only focused on the host or the pathogen responses. For scRNA-Seq, we had also contrasted our results to other datasets that examined gene expression in macrophages infected with *Salmonella* or stimulated with LPS and described commonalities (**Line 371**) and differences in heterogeneity and bimodality (**Line 374; Figure S12**) in the single infected cell responses to these different pathogens. In revising the paper, we have worked to emphasize the new biological insights provided by our approach of sorting subpopulations of infection and performing scRNA-Seq analysis.

Reviewer #2 (Remarks to the Author):

General

The paper describes an interesting and challenging approach to investigate interactions of subpopulations of macrophages and Candida albicans cells on the transcriptional level.

Similar studies have been applied for interactions of bacteria and phagocytes and I completely agree with the authors that the next level of transcriptional profiling studies to study host-pathogen interactions would be to not only looking at the average response of whole populations, but to dissect the response of the different individual subpopulations. From this point of view, the topic is at the forefront of science and of interest for the scientific community.

However, the reason why such studies have yet rarely been published is the fact that it is a very challenging approach with several technical hurdles.

While the expertise of the authors in bioinformatics is obvious and the analysis of the data technical well done, there are a number of critical issues.

(1) A key critical point for all studies of such kind is the cell harvest and RNA extraction protocol. In the current study, the authors harvested the macrophages with/without C. albicans cells and the different C. albicans reporter strains by centrifugation at 37°C for 10 min, then resuspending the samples in PBS/FCS and placing on ice until FACS. The time required for FACS sorting and the conditions for FACS sorting are not defined. Only after that, the samples are frozen until RNA extraction.

Response: The revised manuscript elaborates on the conditions for the FACS sorting (**Pg. 17, Line 549**).

I have strong doubts that the final RNA populations reflect the original transcriptomes of macrophages and C. albicans in their culture medium. Previous studies have shown that fungal cells can change their entire transcriptome within less than 10 min and it is likely that transcriptomes of both macrophages and C. albicans have changed during the procedure described above, including centrifugation and FACS sorting. I strongly believe that an efficient blockage of transcription within a few minutes (eg by freezing or chemicals such as RNALater) is essential. Controls are required to show that the sampling and harvesting protocols do not influence gene expression profiles for a number of representative genes.

Response: The reviewer is correct that we did not add RNALater to cells prior to FACS, as it has been reported to severely decrease GFP signal, as noted in the response to reviewer 1, (see Zaitoun et al. *BMC Res. Notes* 2010). To reduce transcriptional changes not reflective of the experimental conditions, our samples were kept on ice before and after sorting and then flash frozen shortly after collection. As noted in the response to reviewer 1, we have now compared sorted samples at time 0 with unsorted samples at time 0, and found that there is a minor impact on transcriptional changes in macrophages and no impact on *C. albicans*. However, for an abundance of caution regarding this criticism of both reviewer

1 and 2, we have re-analyzed the data to measure differential expression between subpopulations using only sorted samples (see Response 1 to reviewer 1).

(2) I miss a about the dynamics of interactions: at which time point are how many C. albicans cells in contact with macrophages, how many are phagocytosed over the time course, how many fungi are killed, how many produce hyphae, how many escape, how many macrophages are killed etc. I think such a detailed description of the interaction dynamics is essential to understand the profiles and the biological processes in the experimental setting used by the authors (even if that has been done in other studies). This could also show whether the C. albicans cells were all in the same stage when exposed to macrophages after their 37°C incubation step, which is implied when the authors attribute the observed differences on the single-cell level to direct host-fungal cell interactions.

Response: The dynamic nature of *C. albicans* interaction with macrophages has been elegantly demonstrated in previous studies (ie Bain et al *mBio* 2014, Uwamahoro et al *mBio* 2014). This dynamic interaction was also observed during our time lapse microscopy. Specifically, in response to the reviewer's comment, we have included **Figure 1B** in the revised manuscript, which describes the major and specific infecting populations. We were able to use FACS to accurately quantify the interactions. These data indicate that 11% of macrophages had engulfed live *C. albicans* at 10 minutes post exposure and 30% of macrophages had engulfed live *C. albicans* at 4 hours. Additionally, 0% of macrophages contained dead *Candida* at 10 minutes and 1 hour post exposure, 1% of macrophages contained dead *Candida* at 2 hours and 3% of macrophages contained dead at 4 hours post exposure. These results are in line with a previous study (Bain et al *mBio* 2014), which reported that at 30 minutes and 3 hours, respectively, 6% and 21% of macrophages contained live *C. albicans* in the phagosome. We cannot be sure that all *C. albicans* cells were in the same stage when exposed to macrophages, however we included a step to acclimatize the *Candida* to the change from yeast media to RPMI and have updated the text to reflect this (addressed in **Methods section, Lines 537-538**). Our efforts to automate microscopy to analyze these interactions was limited by the pleomorphism exhibited by *C. albicans*, where cell shapes of *Candida* were too irregular to get reliable counts, especially during filamentation.

(3) Although differences between populations are technically described in detail, the key messages are missing, except that different groups of genes linked to different GO terms are differentially expressed. What I found lacking is which new insights we gained from applying this sophisticated technique that we could not obtain by population-based approaches. This is reflected by the rather short discussion, which is also largely technical in nature.

Response: As noted in the response to reviewer 1, we have included new analyses that demonstrate how single cell data can be used to infer infection progression in host and pathogen using asynchronous data and this analysis highlighted genes important for infection fate (**Pg. 10; revised Figure 4C and supplementary S11**). In addition, we have identified genes that appear alternatively spliced in the host at the single cell level and that have differential isoform usages between co-stages of infection (**Pg. 12; new Figure 6**), including genes important for the immune response to *C. albicans* (e.g. Dectin-2). To address the reviewer's concern, we have also substantially modified the discussion (**Pg. 15 and 16**) to make the case for how our method can be best applied, i.e. by comparing to mutants in specific processes and in profiling non-clonal host and pathogen samples.

(4) As such, the study remains descriptive and no clear biological hypothesis has been developed which could be experimentally tested.

Response: The main purpose of this study was to establish methods for dual and single-cell RNA-Seq analysis to study host-fungal pathogen interactions. Additional experiments involving mutants that affect

hyphal formation or survival in macrophages, or comparisons with other *Candida* isolates or species, would help identify specific genes and pathways that could be tested, however this work is beyond the scope of this initial paper. We have modified the discussion to note these future directions (**Pg. 15**).

Further points:

Line 131: "Fewer transcripts were detected in subpopulations of macrophages infected with dead C. albicans (an average of 3,214 host transcripts and 983 fungal transcripts Figure S3A) and...". How meaningful are transcripts of killed C. albicans cells?

Response: While this observation is not important for *C. albicans* transcripts it is meaningful for host transcripts. We have removed this.

Reviewer #3 (Remarks to the Author):

The fundamental question under assessment is an important one – how much variation is there in the interaction between a host immune cell and a pathogen, in this case the opportunistic fungal pathogen C. albicans, and bone marrow-derived macrophages? The approach taken, that of doing RNA seq in isolated cells undergoing the interaction is clearly the next step in an ongoing refinement in assessing the answer to the question, and the authors are clearly competent in the analysis.

I have a number of comments

The abstract is very long for an abstract – efforts should be made to shorten it, focusing on what is really new in the present study and not on results that support previous work.

Response: The abstract has been revised to highlight the major findings of our study.

The introduction does a pretty good job of defining the field and previous work. Remove the comma on line 77.

Response: The comma has been removed.

The ability to sort the interaction sets is critical to the study – I would like to see the S2A data presented in the manuscript and not in the supplementary data.

Response: Figure S2A has been moved to the main manuscript (**revised Figure 1B**).

In lines 124/125 it is claimed that increases in numbers of engulfed Candida cells represents further engulfment. Is it possible that some of the increase represents Candida cells proliferating in the engulfed state – and if so do these cells represent yet another possible class of host-pathogen interactions?

Response: The reviewer is correct. Microscopically, we and others (Bain et al *mBio* 2014; Uwamahoro et al *mBio* 2014; Erwig and Gow *Nat. Rev. Microbiol.* 2016) have observed that macrophages are capable of phagocytosing multiple *Candida*. This would likely alter the magnitude of the fluorescent signal. However, we acknowledge that cell division is possible and now describe this an alternative possibility (**Lines 26-27 in Supplementary notes**).

In lines 130/131 the authors note that the class of macrophage that have engulfed and potentially killed the Candida cells provided very poor data, and are essentially dropped from the analysis, but no explanation is provided. This is a clearly a central class to examine for the study and needs a better fate.

Why should this data set be poor? This is a critical point – the authors should either provide this data or provide a good explanation for the failure.

Response: Here we need to clarify that macrophage data from this subpopulation was included previously in the section “*Subpopulations of macrophages showed major pathogen recognition and pro-inflammatory response to C. albicans and shift profiles at late time course*” (Pg. 7). We analyzed this subpopulation separately, since substantially fewer transcripts were detected compared to other sorted populations (< 3,214, **Figures S1A, S6**) and in addition these samples had lower correlation of biological replicates (e.g. Pearson’s $r < 0.56$). This lower coverage was expected for this subpopulation, as only up to 3% of each sample collected at each time point was comprised of macrophages infected with dead *C. albicans* (**revised Figure 1B**). We have explained this in the manuscript (Pg. 7), included a new Supplementary Notes with these details and highlight this would be an important parameter to consider in further experiments.

Line 152 - whereas as one word

Response: This has been fixed in the revised manuscript.

The initial studies represent analyses of pooled samples selected by the FACS sorting, and these results fundamentally support previous studies directed at the pathogen and host cell responses. Lines 181-185 provide evidence for an up-regulated set that is not attributed to phagocytosis – providing a possible signal here would be useful.

Response: The differential expression pattern between these two subpopulations across this time course clarifies which pathways and processes reported in previous studies are specific to phagocytosed *C. albicans* and determined the set of genes specific of this response. Comparing exposed and phagocytosed fungal stages, genes were largely down-regulated in live, phagocytosed *C. albicans* at 1 and 2 hours. This was established as expression levels of these genes largely did not change in exposed *C. albicans* over the time course and recovered their expression levels by 4 hours in phagocytosed *C. albicans*. We did not find genes highly induced in un-engulfed *Candida* relative to phagocytosed cells. We have clarified this text (**Pg. 5; Line 154**).

Lines 189/190 suggest that it is the presence of macrophages and not just the medium that is triggering hyphal development, but this claim is poorly supported.

Response: We have modified the text to note that this could also be due to the media (**Pg. 5; Line 154**)

Line 193 – what explains a class of genes that are repressed by interaction and induced by phagocytosis? This is an interesting (confusing) behaviour and deserves a comment.

Response: After we re-analyzed this data set using only sorted samples (see responses to reviewers 1 and 2 above), this set of genes no longer appeared significantly differentially expressed.

Next the authors investigate single cell response. This is the critical new area for the paper – what is found in single cells that improves our understanding derived from pooled cells? A first point made is that individual cells have more variation than seen in the previous multi-cell analyses, and then the authors note that pooling the data from the individual cells reduces variation. This seems self evident – what could prevent the pooled samples reducing variation?

Response: We have revised the manuscript to highlight what additional information can be gleaned from single cell analysis. In addition, as noted in the response to reviewer 1, we have included new analyses

that demonstrate how single cell data can be used to infer the trajectory of host and pathogen interactions from asynchronous data and highlighted genes important for infection fate (**Pg. 10; new Figure 4C and supplementary S11**). In addition, we have added new analysis that identified genes that appear alternatively spliced at the single cell level, and that have differential exon usages between macrophage co-stages of infection (**Pg. 12; new Figure 6**), including genes important for the immune response to *C. albicans* (e.g. Dectin-2). To address the reviewer's concern we have also substantially modified the discussion (**Pg. 15 and 16**) to make the case for how our method can be best applied, *i.e.* profiling the expression of mutants in specific processes and in non-clonal host and pathogen samples. The main point of the analysis examining the pooled single-cells was to confirm that they show high correlation with the population level samples, validating the quality of the single cell data. Since this is the first, host-fungal pathogen single-cell approach, we do consider this quality check important to establish that this system can accurately detect gene expression in single infected macrophages and phagocytosed *C. albicans*.

The key data that come from the ability to analyze individual cells is the suggestion that there is a coupled response in the cells – a bimodal relationship in the pathogen linked to a bimodal response in the immune cells. In particular – induction of hyphae is coupled to down-regulation of the pro-inflammatory state of the phagocytes. This is an exciting observation, but needs some context. The authors themselves note that the driver of this correlation is not known “it is unclear whether expression heterogeneity among individual infected macrophages results from or results in expression bimodality among phagocytosed C. albicans.” Perhaps doing the experiment with a non-hyphal C. albicans mutant would clarify this.

Response: To address the reviewer's point, we have carried out a finer scale analysis of the single infected macrophages and phagocytosed *C. albicans* pairs using Monocle. This characterized trajectories of host-pathogen cells across our two time points, and this data suggests that there is an earlier shift to co-stage 2 for *Candida* cells (**revised Figure 4C**). We found that the fraction of phagocytosed *C. albicans* shifting to high expression of filamentation genes across ordered pseudo-time increased slightly faster than the fraction of macrophages in each group decreasing the levels of expression of pro-inflammatory cytokines (10% more cells in the late pseudo-time range; **revised Figure 4C**), suggesting that the expression heterogeneity in macrophages could be driven by the expression heterogeneity in *C. albicans* (**Pg. 11; Line 319**). This is in line with other studies that showed that *Candida* filamentation could occur inside the phagosome and induce macrophages expulsion, pyroptosis and cell damage (**Discussion; Pg. 15**). We agree with the reviewer that carrying out experiments with *Candida* mutated for hyphal formation and other processes is a logical next step for a follow up study.

REVIEWERS' COMMENTS:

Reviewer #1 (Remarks to the Author):

This is a fantastically improved manuscript.
The authors have adequately addressed all of my comments.
Really nice work!

Reviewer #2 (Remarks to the Author):

The revised manuscript has clearly increased in the overall quality of the paper and the relevance of the presented data.

However, there are still critical points left:

(1) The potential effect of sorting has been partially solved by the additional comparison of sorted versus non-sorted cells and by showing that FACS sorting appears to not have a large impact on the transcriptional profile (response to reviewer 1). Nevertheless, I suggest that the authors at least indicate the overall time required for FACS sorting to give the reader an impression to which environmental conditions both macrophages and *C. albicans* cells are exposed (for how long) prior to RNA extraction.

Similarly, the authors should comment on the influence of 10 min incubation in TrypLE and 10 min centrifugation at 300 g prior to FACS sorting. Although this may not influence the observed overall differences between the different populations of macrophages and *C. albicans* (since all are treated similarly), it very likely has an influence on the expression profiles. For example, when looking at Fig. 4b (and d) it seems as there are already significant differences between *Candida*-exposed macrophages versus non-exposed macrophages at time point 0 h. This indicates either that the response of macrophages is extremely quick and/or that the time point 0 h is not a real time point 0 h and macrophages have adapted to something else.

This should be at least critically discussed.

As indicated in my previous comments, I see this work at the forefront of science in the field of mycology and infection biology, but the reader should also be informed about the weaknesses. In this context, it is also likely that the Deep Red membrane stain of the macrophages has an influence on the activities of the macrophages. It would be interesting to know how this may influence phagocytosis rates (but not essential for this manuscript).

(2) The authors have now provided a more detailed picture about the dynamics of interactions and have used FACS to quantify the interactions. These data indicate that only a minority of macrophages has actually engulfed *C. albicans* cells and only a very small fraction has been able to kill fungi even at time point 4 h. This indicates that in the used experimental setting, like in most other similar studies of *C. albicans* – macrophage interactions, the fungus is actually killing the macrophages and not vice versa, which should influence the interpretation of the data. This view is also supported by the observation that phagocytosed *C. albicans* cells show strong expression of hyphal-associated genes (increasing with time) and thus proliferate (see reviewer 3). Macrophages unable to block hypha will eventually be killed as shown in a number of reports. The observed transcriptional response of macrophages, which had phagocytosed *C. albicans* cells, at later time points may thus reflect the response of dying cells.

Alternatively, could it be that the down-regulation (line 466) is not really dependent upon *C. albicans* gene expression, but rather an intrinsic regulation, a self-limiting reaction of the macrophages to avoid immune overreaction? With other words, the correlation may not imply a causation.

(3) Finally, I still miss a biological key message. The biological messages highlighted in the abstract are known facts from previous studies. Nevertheless, the authors have stressed that the main purpose of the study was to establish methods for dual and single-cell RNA-Seq analysis. This aim was certainly successfully reached and I appreciate the effort of the authors to establish a single cell profiling approach in this context, which may lead to further novel insights into fungal/host interactions.

Minor:

line 152 -- "during at 4 hours upon phagocytosis"

line 268 -- fewer transcripts due to smaller transcriptome, but also simply size? Obviously, larger cells need to produce and maintain higher amounts of RNA and protein to sustain biomass and function, although the genome content often remains constant (eg Marguerat et al. Trends Genet. 2012).

Reviewer #3 (Remarks to the Author):

In general the authors have done a good job with my concerns. I have a couple of minor comments to improve the legibility of the presentation that can be considered.

In parallel with the analysis of *C. albicans* gene expression, we also examined the transcriptional response of macrophages. Across all samples, we identified 577 DEGs (FC > 4; FDR < 0.001; Data set 2), which grouped into four clusters with similar expression patterns (Figures 2C, 2D).

The labelling of this figure appears mixed up? I think the authors mean for example, on line 122 Figure 2B, but this is not the only problem. I guess they mean 2c on line 137. Make sure the connections are correct.

Figure 2B, perhaps they have captured a signature of M0 in the presence of Ca, but it isn't totally convincing that it is a signature of "infection". If there is something significantly different between the exposed and infected halves of the figure, they would need to dig into that a bit deeper. I guess that previous bulk profiling found these specific pathways for Ca captured in the phagosome. I guess I'm confused though - the pattern seems to be the same for exposed (but not infected; left) vs exposed (and infected; right). Perhaps this could be highlighted a bit better.

In general, Figure 2B is used to support a lot of statements in the paper. For example, that there are ZCTFs identified but they are not labelled in the figure.

For Figure 2a, the internalized timepoint 4 seems basically overexpression everywhere. Actually the authors say in multiple places in the paper that the signature is rapidly expressed at early time points but then repressed at later time points. I don't see that at all. This appears at odds with their statement on line 137

The labelling of cluster 1 in Figure 2C (correct labelling) seems at odds with their text on lines 140, 141.

The points being made in lines 301-304 could be clarified.

Similarly, the Monocle analysis lines 320-340 could be more clear.

I am not sure why the paragraph at line 397 is placed after the BRIE discussion and not directly beside the previous DD discussion. they seem to be similar in nature.

These points are mainly suggestions for points that could aid the clarity of the presentation.

REVIEWERS' COMMENTS:

Reviewer #1 (Remarks to the Author):

This is a fantastically improved manuscript.
The authors have adequately addressed all of my comments.
Really nice work!

Response: We appreciate the reviewer's feedback on our revisions.

Reviewer #2 (Remarks to the Author):

The revised manuscript has clearly increased in the overall quality of the paper and the relevance of the presented data.

However, there are still critical points left:

(1) The potential effect of sorting has been partially solved by the additional comparison of sorted versus non-sorted cells and by showing that FACS sorting appears to not have a large impact on the transcriptional profile (response to reviewer 1). Nevertheless, I suggest that the authors at least indicate the overall time required for FACS sorting to give the reader an impression to which environmental conditions both macrophages and *C. albicans* cells are exposed (for how long) prior to RNA extraction.

Similarly, the authors should commend on the influence of 10 min incubation in TrypLE and 10 min centrifugation at 300 g prior to FACS sorting. Although this may not influence the observed overall differences between the different populations of macrophages and *C. albicans* (since all are treated similarly), it very likely has an influence on the expression profiles. For example, when looking at Fig. 4b (and d) it seems as there are already significant differences between *Candida*-exposed macrophages versus non-exposed macrophages at time point 0 h. This indicates either that the response of macrophages is extremely quick and/or that the time point 0 h is not a real time point 0 h and macrophages have adapted to something else.

This should be at least critically discussed.

Response: We have added additional details to the Methods description of the FACS sorting including additional estimates of the incubation time with TrypLE, centrifugation, and the time for the sorting of each sample. To address the reviewer's concern, we have noted in the results that the processing time may have affected the results and that we compared only sorted samples to control for this.

As indicated in my previous comments, I see this work at the forefront of science in the field of mycology and infection biology, but the reader should also be informed about the weaknesses. In this context, it is also likely that the Deep Red membrane stain of the macrophages has an influence on the activities of the macrophages. It would be interesting to know how this may influence phagocytosis rates (but not essential for this manuscript).

Response: We have not evaluated how the Deep Red stain may impact macrophage activities, but will keep the reviewer's point in mind for future work.

(2) The authors have now provided a more detailed picture about the dynamics of interactions and have used FACS to quantify the interactions. These data indicate that only a minority of macrophages has actually engulfed *C. albicans* cells and only a very small fraction has been able to kill fungi even at time point 4 h. This indicates that in the used experimental setting, like

in most other similar studies of *C. albicans* – macrophage interactions, the fungus is actually killing the macrophages and not vice versa, which should influence the interpretation of the data. This view is also supported by the observation that phagocytosed *C. albicans* cells show strong expression of hyphal-associated genes (increasing with time) and thus proliferate (see reviewer 3). Macrophages unable to block hypha will eventually be killed as shown in a number of reports. The observed transcriptional response of macrophages, which had phagocytosed *C. albicans* cells, at later time points may thus reflect the response of dying cells.

Alternatively, could it be that the down-regulation (line 466) is not really dependent upon *C. albicans* gene expression, but rather an intrinsic regulation, a self-limiting reaction of the macrophages to avoid immune overreaction? With other words, the correlation may not imply a causation.

Response: We have now noted this alternatively possibility suggested by the reviewer in the text.

(3) Finally, I still miss a biological key message. The biological messages highlighted in the abstract are known facts from previous studies. Nevertheless, the authors have stressed that the main purpose of the study was to establish methods for dual and single-cell RNA-Seq analysis. This aim was certainly successfully reached and I appreciate the effort of the authors to establish a single cell profiling approach in this context, which may lead to further novel insights into fungal/host interactions.

Response: The large scale view of transcriptional responses that we detect are in line with previous studies, and this is validation of our approach. By sorting cells and using scRNA-Seq, we can now separate these changes to occur in stages, ie the early response of macrophages to *Candida* exposure and the synchronized shift in expression at 4 hours that our pseudo time analysis suggests originates with a shift in fungal gene expression. The analyses of bimodality and differential splicing also help clarify the variation that exists even within a clonal population of cells, and is important to consider in any 'bulk' approaches.

Minor:

line 152 -- "during at 4 hours upon phagocytosis"

Response: The text was edited to read "Genes most strongly induced at 4 hours upon phagocytosis".

line 268 -- fewer transcripts due to smaller transcriptome, but also simply size?

Obviously, larger cells need to produce and maintain higher amounts of RNA and protein to sustain biomass and function, although the genome content often remains constant (eg Marguerat et al. Trends Genet. 2012).

Response: We have added text to note that cell size also likely plays a role in relative transcript abundance.

Reviewer #3 (Remarks to the Author):

In general the authors have done a good job with my concerns. I have a couple of minor comments to improve the legibility of the presentation that can be considered.

In parallel with the analysis of *C. albicans* gene expression, we also examined the transcriptional response of macrophages. Across all samples, we identified 577 DEGs (FC > 4; FDR < 0.001; Data set 2), which grouped into four clusters with similar expression patterns (Figures 2C, 2D).

The labelling of this figure appears mixed up? I think the authors mean for example, on line 122 Figure 2B, but this is not the only problem. I guess they mean 2c on line 137. Make sure the connections are correct.

Response: We apologize for these errors in how the figure was called out in the text; the text was updated to swap the call outs for Figure 2B and 2C.

Figure 2B, perhaps they have captured a signature of M0 in the presence of Ca, but it isn't totally convincing that it is a signature of "infection". If there is something significantly different between the exposed and infected halves of the figure, they would need to dig into that a bit deeper. I guess that previous bulk profiling found these specific pathways for Ca captured in the phagosome. I guess I'm confused though - the pattern seems to be the same for exposed (but not infected; left) vs exposed (and infected; right). Perhaps this could be highlighted a bit better.

Response: We have reported that exposed and infected macrophages have similar transcriptional responses based on principal component analysis (see line 121; Figure 2B) and differentially expressed genes (line 195; Figure 2D), and that the major variation was over time, rather than between exposed or infected macrophage subpopulations. To address comment we have added text noting two pathways (pathogen recognition and ROS production) that were differentially regulated between exposed and infected cells.

In general, Figure 2B is used to support a lot of statements in the paper. For example, that there are ZCTFs identified but they are not labelled in the figure.

Response: The Zinc cluster transcription factors were not detected by GO enrichment so are not summarized in Figure 2C (and in Supplementary Table 3); they are listed in Supplementary Table 2 and we have added a new call out to this table where these genes are noted in the text.

For Figure 2a, the internalized timepoint 4 seems basically overexpression everywhere. Actually the authors say in multiple places in the paper that the signature is rapidly expressed at early time points but then repressed at later time points. I don't see that at all. This appears at odds with their statement on line 137

Response: The differential expression of internalized time point 4 needs to be in comparison with other time points and conditions. Higher induction at this time point is clearly depicted in cluster 1 (Figure 2C). Repression of genes at 1 and 2 hours is depicted in cluster 3 and 4 (Figure 2C).

The labelling of cluster 1 in Figure 2C (correct labelling) seems at odds with their text on lines 140, 141.

Response: We have revised the text to refer to Figure 2C.

The points being made in lines 301-304 could be clarified.

Response: We have revised the text to clarify the comparison here between the expression profiles of the single cells and sorted samples.

Similarly, the Monocle analysis lines 320-340 could be more clear.

Response: We have added an additional introductory sentence describing the goal of the Monocle analysis, and have added detail linking the co-states to time points and noting the earlier transition of *Candida*.

I am not sure why the paragraph at line 397 is placed after the BRIE discussion and not directly beside the previous DD discussion. they seem to be similar in nature.

Response: The BRIE analysis was only carried out for macrophage transcripts, so we feel that is best to follow the discussion of macrophage bimodality.

These points are mainly suggestions for points that could aid the clarity of the presentation.

Response: We appreciate the reviewer's feedback and careful review of our paper.